# Not all feldspars are equal: a survey of ice nucleating properties across the feldspar group of minerals

**Alexander D. Harrison[1‡], Thomas F. Whale[1‡*], Michael A. Carpenter[2], Mark A. Holden[1], Lesley Neve[1], Daniel O'Sullivan[1], Jesus Vergara Temprado[1], Benjamin J. Murray[1*]**

[1]School of Earth and Environment, University of Leeds, Leeds, LS2 9JT, UK

[2] Department of Earth Sciences, University of Cambridge, Downing Street, Cambridge CB2 3EQ, UK

Correspondence to: T. F. Whale (t.f.whale@leeds.ac.uk) and B. J. Murray (b.j.murray@leeds.ac.uk)

[‡] Both authors contributed equally to the paper

## Abstract

Mineral dust particles from wind-blown soils are known to act as effective ice nucleating particles in the atmosphere and are thought to play an important role in the glaciation of mixed phase clouds. Recent work suggests that feldspars are the most efficient nucleators of the minerals commonly present in atmospheric mineral dust. However, the feldspar group of minerals is complex, encompassing a range of chemical compositions and crystal structures. To further investigate the ice-nucleating properties of the feldspar group we measured the ice nucleation activities of 15 characterised feldspar samples. We show that alkali feldspars, in particular the potassium feldspars, generally nucleate ice more efficiently than feldspars in the plagioclase series which contain significant amounts of calcium. We also find that there is variability in ice nucleating ability within these groups. While five out of six potassium-rich feldspars have a similar ice nucleating ability, one potassium rich feldspar sample and one sodium-rich feldspar sample were significantly more active. The hyper-active Na-feldspar was found to lose activity with time suspended in water with a decrease in mean freezing temperature of about 16°C over 16 months; the mean freezing temperature of the hyper-active K-feldspar decreased by 2°C over 16 months, whereas the 'standard' K-feldspar did

not change activity within the uncertainty of the experiment. These results, in combination with a review of the available literature data, are consistent with the previous findings that potassium feldspars are important components of arid or fertile soil dusts for ice nucleation. However, we also show that there is the possibility that some alkali feldspars may have enhanced ice nucleating abilities, which could have implications for prediction of ice nucleating particle concentrations in the atmosphere.

## 1 Introduction

Clouds containing supercooled liquid water play an important role in our planet's climate and hydrological cycle, but the formation of ice in these clouds remains poorly understood (Hoose and Möhler, 2012). Cloud droplets can supercool to below -35°C in the absence of particles capable of nucleating ice (Riechers et al., 2013;Herbert et al., 2015), hence clouds are sensitive to the presence of ice nucleating particles (INPs). A variety of aerosol types have been identified as INPs (Murray et al., 2012;Hoose and Möhler, 2012) , but mineral dusts from deserts are thought to be important INPs over much of the globe and in a variety of cloud types (DeMott et al., 2003;Hoose et al., 2008;Hoose et al., 2010;Niemand et al., 2012;Atkinson et al., 2013).

Atmospheric mineral dusts are composed of weathered mineral particles from rocks and soils, and are predominantly emitted to the atmosphere in arid regions such as the Sahara (Ginoux et al., 2012). The composition and relative concentrations of dust varies spatially and temporally but it is generally made up of only a handful of dominant minerals. The most common components of dust reflect the composition of the continental crust and soil cover, with clay minerals, feldspars and quartz being major constituents. Until recently, major emphasis for research has been placed on the most common minerals in transported atmospheric dusts, the clays. It has now been shown that, when immersed in water, the feldspar component nucleates ice much more efficiently than the other main minerals that make up typical desert dust (Atkinson et al., 2013;Augustin-Bauditz et al., 2014;O'Sullivan et al., 2014;Niedermeier et al., 2015;Zolles et al., 2015). This is an important finding as it has been demonstrated that feldspar is a common component of aerosolised mineral dusts (Glaccum and Prospero, 1980;Kandler et al., 2009;Kandler et al., 2011;Atkinson et al., 2013;Perlwitz et al., 2015). Feldspar particles in the atmosphere tend to be larger than clay particles and so will have shorter lifetimes in the atmosphere, however aerosol modelling

work has suggested that feldspar particles can account for many observations of INP concentrations around the world (Atkinson et al., 2013). Work conducted below water saturation using a continuous flow diffusion chamber has also concluded that feldspars, particularly orthoclase feldspars, nucleate ice at lower relative humidity in the deposition mode than other common dust minerals (Yakobi-Hancock et al., 2013). While all available evidence indicates that feldspars are very effective INPs, it must also be recognised that feldspars are a group of minerals with differing compositions and crystal structures. Therefore, in this study we examine immersion mode ice nucleation by a range of feldspar samples under conditions pertinent to mixed phase clouds.

An additional motivation is that determining the nature of nucleation sites is of significant fundamental mechanistic interest and is likely to help with further understanding of ice nucleation in the atmosphere (Vali, 2014;Freedman, 2015;Slater et al., 2015). By characterising a range of feldspars and associating them with differences in ice nucleation activity it might be possible to build understanding of the ice nucleation sites on feldspars. Some work has been conducted in this area already. Augustin-Bauditz et al. (2014) concluded that microcline nucleates ice more efficiently than orthoclase on the basis of ice nucleation results looking at a microcline feldspar and several mixed dusts. Zolles et al. (2015) recently found that a plagioclase and an albite feldspar nucleated ice less well than a potassium feldspar and suggested that the difference in the ice nucleation activity of these feldspars is related to the difference in ionic radii of the cations and the local chemical configuration at the surface. They suggested that only potassium feldspar will nucleate ice efficiently because the $K^+$ is kosmotropic (structure making) in the water hydration shell while $Ca^{2+}$ and $Na^+$ are chaotropic (structure breaking).

There has been much interest in the study of ice nucleation using molecular dynamics simulations (e.g. (Hu and Michaelides, 2007;Cox et al., 2012;Reinhardt and Doye, 2014;Lupi and Molinero, 2014;Lupi et al., 2014;Zielke et al., 2015;Cox et al., 2015a, b;Fitzner et al., 2015). To date there has been little overlap between work of this nature and laboratory experiments. This has been due to difficulties in conducting experiments on similar timescales and spatial extents between real-world and computational systems. While these obstacles are likely to remain in place for some time, the feldspars may offer the opportunity to address this deficit by providing qualitative corroboration between computational and laboratory results. For instance, it may be possible to study ice nucleation on different types of feldspar computationally. If differences in nucleation rate observed also occur in the

laboratory greater weight may be placed on mechanisms determined by such studies and so a
mechanistic understanding of ice nucleation may be built up.
In this paper we have surveyed 15 feldspar samples with varying composition for their ice
nucleating ability in the immersion mode. It will be shown that feldspars rich in alkali metal
cations tend to be much better at nucleating ice than those rich in calcium.  First, we
introduce the feldspar group of minerals.

## 9   2        The feldspar group of minerals

The feldspars are tectosilicates (also called framework silicates) with a general formula of
$XAl(Si,Al)Si_2O_8$, where X is usually potassium, sodium or calcium (Deer et al., 1992).
Unlike clays, which are phyllosilicates (or sheet silicates), tectosilicates are made up of three
dimensional frameworks of silica tetrahedra. Substitution of Si with Al in the structure is
charge balanced by cation addition or replacement within the cavities in the framework. This
leads to a large variability of composition in the feldspars and means that most feldspars in
rocks have compositions between end-members of sodium-, calcium- or potassium-feldspars
(Deer et al., 1992;Wenk and Bulakh, 2004). A ternary representation of feldspar
compositions is shown in Figure 1. All feldspars have very similar crystal structures, but the
presence of different ions and degrees of disorder related to the conditions under which they
crystallised from the melt (lava or magma) yields subtle differences which can result in
differing symmetry.
There are three polymorphs (minerals with the same composition, but different crystal
structure) of the potassium end-member, which are microcline, orthoclase and sanidine. The
polymorphs become more disordered in terms of Al placement in the tetrahedra from
microcline to sanidine, respectively. The structures of feldspars which form from a melt vary
according to their cooling rate. If cooling is fast (volcanic), sanidine is preserved. If cooling is
slow, in some granites for example, microcline may be formed. Feldspars formed in
metamorphic rocks have high degrees of Al/Si order. The sodium end-member of the
feldspars is albite and the calcium end-member is anorthite. Feldspars with compositions
between sodium and calcium form a solid solution and are collectively termed the plagioclase
feldspars with specific names for different composition ranges. Feldspars between sodium

and potassium end-members are collectively termed the alkali feldspars and can be structurally complex. A solid solution series exists between high albite and sanidine ('high' refers to high temperature character which is preserved on fast cooling), but not between low albite and microcline ('low' refers to low temperature character which is indicative of slow cooling rates). In contrast to the series between sodium and calcium, and sodium and potassium, there are no feldspars between calcium and potassium end-members because calcium and potassium ions do not actively substitute for one another within the framework lattice due their difference in size and ionic charge (Deer et al., 1992;Wenk and Bulakh, 2004).

There is limited information about the composition of airborne atmospheric mineral dusts (Glaccum and Prospero, 1980;Kandler et al., 2007;Kandler et al., 2009); where mineralogy is reported the breakdown of the feldspar family has only been done in a limited way. Atkinson et al. (2013) compiled the available measurements and grouped them into K-feldspars and plagioclase feldspars (see the Supplementary Table 1 in Atkinson et al. (2013)). This compilation indicates that the feldspar type is highly variable in atmospheric dusts, with K-feldspars ranging from 1 to 25% by mass (with a mean of 5%) and plagioclase feldspars ranging from 1 to 14% (with a mean of 7%). The feldspar component of airborne dusts is highly variable and the nucleating ability of the various components needs to be investigated.

In order to aid the discussion and representation of the data we have grouped the feldspars into three groups: the plagioclase feldspars (not including albite), albite (the sodium rich corner of the ternary diagram) and potassium (K-) feldspars (microcline, sanidine and orthoclase). The K-feldspars contain varying amounts of sodium, but their naming is determined by their crystal structure. We also collectively refer to albite and potassium feldspars as alkali feldspars.

## 3    Samples and sample preparation

A total of 15 feldspars were sourced for this study. Details of the plagioclase feldspars tested are in Table 1 and details of the alkali feldspars are in Table 2. We have made use of a series of characterised plagioclase feldspars which were assembled for previous studies (Carpenter et al., 1985;Carpenter, 1986;Carpenter, 1991). The other samples were sourced from a range of repositories, detailed in Tables 1 and 2. The naming convention we have used in this paper

is to state the identifier of the specific sample followed by the mineral name. For example, BCS 376 microcline is a microcline sample from the Bureau of Analysed Samples with sample code 376. In other cases, such as Amelia Albite, the sample is from a traceable source and is commonly referred to with this name and when a code is used, such as 97490 plagioclase, the code links to the cited publications.

Anorthite glass and synthetic anorthite ANC 68 were tested for ice nucleating efficiency to investigate the impact of crystal structure in feldspar. The anorthite glass was produced by Carpenter (1991) by melting natural calcite with reagent grade $SiO_2$ and $Al_2O_3$ at 1680°C for 3 hours. The melt was then stirred before air cooling. The resulting glass was then annealed at 800°C to relieve internal stresses. The composition of the resulting glass was shown to be stoichiometric $CaAl_2Si_2O_8$. Synthetic anorthite ANC 68 was produced by heating a sample of this glass to 1400°C for 170 hours. As these two samples are chemically identical, differing only in that one is amorphous and the other crystalline, comparison of the ice nucleating efficiency of the two samples has the potential to reveal information about the impact of feldspar crystal structure on ice nucleating efficiency. Feldspars 148559, 21704a, 67796b and 97490 plagioclase and Amelia albite are natural samples that form a solid solution series covering the plagioclase series from nearly pure anorthite to nearly pure albite as seen in Table 1.

The alkali feldspars used here have not previously been characterised. Rietveld refinement of powder XRD patterns was carried out using TOtal Pattern Analysis Solutions (TOPAS) to determine the phase of the feldspar present. The results of this process are presented in Table 2. The surface areas of all the feldspars were measured by Brunauer-Emmett-Teller (BET) nitrogen gas adsorption (see Sect. 4). All samples, unless otherwise stated, were ground to reduce the particle size and increase the specific surface area using a mortar and pestle which were scrubbed with pure quartz then cleaned with deionised water and methanol before use. Grinding of most samples was necessary in order to make the particles small enough for our experiments. Amelia albite was the only material tested both in an unground state (or at least not a freshly ground state) and a freshly ground state. Suspensions of known concentration were made up gravimetrically using Milli-Q water (18.2 MΩ.cm). Except where stated otherwise the suspensions were then mixed for a few minutes using magnetic stirrers prior to use in ice nucleation experiments.

The activity of Amelia albite and TUD #3 microcline decrease during repeat experiments
conducted approximately 30 minutes after initial experiments. Based on this observation
three samples, the BCS 376 microcline, ground Amelia albite and TUD #3 microcline, were
tested for changes in ice nucleating efficiency with time, when left in suspension at room
temperature. BCS 376 microcline was chosen as it has been previously studied and the
activity decrease was not seen on the time scale of ~ 30 minutes previously. Ice nucleation
efficiency was quantified at intervals over 11 days. Between experiments the suspensions
were left at room temperature without stirring and then stirred to re-suspend the particulates
for the ice nucleation experiments. Suspensions of the three dusts were also tested 16 months
after initial experiments were performed to determine the long term impact of contact with
water on ice nucleation efficiency.

## 4        Experimental method and data analysis

In order to quantify the efficiency with which a range of feldspar dusts nucleate ice we made
use of the microliter Nucleation by Immersed Particle Instrument (µl-NIPI). This system has
been used to make numerous ice nucleation measurements in the past  and has been described
in detail by Whale et al. (2015). Briefly, $1 \pm 0.025$ µl droplets of an aqueous suspension,
containing a known mass concentration of feldspar particles are pipetted onto a hydrophobic
coated glass slide. This slide is placed on a temperature controlled stage and cooled from
room temperature at a rate of 5 °C min$^{-1}$ to 0 °C and then at 1 °C min$^{-1}$ until all droplets are
frozen.  Dry nitrogen is flowed over the droplets at 0.2 l min$^{-1}$ to prevent frozen droplets from
affecting neighbouring liquid droplets, droplets evaporate slowly during experiments
however this has been shown to have no detectable effect on freezing temperatures (Whale et
al., 2015). Whale et al. (2015) demonstrated that a dry nitrogen flow prevents condensation
and frost accumulating on the glass slide so ice from a frozen droplet cannot trigger freezing
in neighbouring droplets. Freezing is observed with a digital camera, allowing determination
of the fraction of droplets frozen as a function of temperature. Multiple experiments have
been combined to produce single sets of data for each mineral. Suspensions of the feldspars
were made up gravimetrically and specific surface areas of the samples were measured using
the Brunauer–Emmett–Teller (BET) $N_2$ adsorption method using a Micromeritics TriStar
3000. Here the µl-NIPI technique is used to for immersion mode nucleation experiments.
To allow comparison of the ability of different materials to nucleate ice, the number of active
sites is normalised to the surface area available for nucleation. This yields the ice nucleation
active site density, $n_s(T)$. $n_s(T)$ is the number of ice nucleating sites that become active per
surface area on cooling from 0°C to temperature $T$ and can be calculated using (Connolly et
al., 2009):
$$\frac{n(T)}{N} = 1 - \exp(-n_s(T)A) \tag{1}$$
Where $n(T)$ is the number of droplets frozen at temperature $T$, $N$ is the total number of
droplets in the experiment and $A$ is the surface area of nucleator per droplet.
Active sites may be related to imperfections in a crystal structure, such as cracks or defects,
or may be related to the presence of quantities of other more active materials located in
specific locations at a surface.  While the fundamental nature of active sites is not clear, and
may be different for different materials, $n_s$ is a pragmatic parameter which allows us to
empirically define the ice nucleating efficiency of a range of materials (Vali, 2014).
This description is site specific and does not include time dependence. The role of time
dependence in ice nucleation has recently been extensively discussed (Vali, 2014;Vali et al.,
2014;Vali, 2008;Herbert et al., 2014;Wright et al., 2013). For feldspar (at least for BCS 376
microcline) it is thought that the time dependence of nucleation is relatively weak and that the
particle to particle, or active site to active site, variability is much more important (Herbert et
al., 2014). The implication of this is that specific sites on the surface of most nucleators,
including feldspars, nucleate ice more efficiently than the majority of the surface. As this
study is aimed at comparing and assessing the relative ice nucleating abilities of different
feldspars we have not determined the time dependence of observed ice nucleation in this
work, although this would be an interesting topic for future study.
By assuming that the BET surface area of the feldspar powders is made up of monodisperse
particles it can be estimated that droplets containing 1 wt% of feldspar will each contain
around $10^6$ particles. While there will be a distribution of particle sizes we assume that there
are enough particles per droplet that the uncertainty in surface area per droplet due to the
distribution of particles through the droplets is negligible. In contrast, it has been suggested
that ice nucleation data could be explained by variability of nucleator surface area through the
droplet population (Alpert and Knopf, 2016).  Our assumption that each droplet contains a
representative surface area is supported by our previous work where we show that $n_s$ derived

from experiments with a range of feldspar concentrations are consistent with one another (Whale et al., 2015;Atkinson et al., 2013). If the particles were distributed through the droplets in such a way that some droplet contained a much larger surface area of feldspar than others we would expect the slope of $n_s$ with temperature to be artificially shallow. The slope would be artificially shallow because droplets containing more than the average feldspar surface area would tend to freeze at higher temperatures and vice versa. However, the fact that $n_s$ data for droplets made from suspensions made up with a wide range of different feldspar concentrations all line up shows that the droplet to droplet variability in feldspar surface area is minor (Atkinson et al., 2013;Whale et al., 2015). Hence, the droplet to droplet variability in feldspar surface area is neglected and the uncertainty in surface area per droplet in these experiments is estimated from the uncertainties in weighing, pipetting and specific surface area of the feldspars. Indeed, Murray et al (2011) found that even with picolitre droplets containing 10's of particles per droplet median nucleation temperatures scaled well with surface area per droplet calculated in the way used in this work.

In order to estimate the uncertainty in $n_s(T)$ due to the randomness of the distribution of the active sites in droplet freezing experiments, we conducted Monte Carlo simulations. Wright and Petters (2013) previously adopted a similar approach to simulate the distribution of active sites in droplet freezing experiments.  In these simulations, we generate a list of possible values for the number of active sites per droplet ($\mu$). The theoretical relationship between the fraction of droplets frozen and $\mu$ can be derived from the Poisson distribution:

$$\frac{n(T)}{N} = 1 - \exp(-\mu) \tag{2}$$

The simulation works in the following manner. First, we take a value of $\mu$ and we simulate a corresponding random distribution of active sites through the droplet population for an experiment. Every droplet containing one or more active sites is then considered to be frozen. In this way, we can obtain a simulated value of the fraction frozen for a certain value of $\mu$. Repeating this process many times and for all the possible values of $\mu$, we obtain a distribution of possible values of $\mu$ that can explain each value of the observed fraction frozen. This resulting distribution is neither Gaussian nor symmetric, so in order to propagate the uncertainty to $n_s(T)$ values, we take the following steps. First, we generate random values of $\mu$ following the corresponding previously simulated distribution for each value of the fraction frozen. Then, we simulate random values of $A$ following a Gaussian distribution centred on the value derived from the specific surface area per droplet with the standard

deviation derived from the uncertainty in droplet volume and specific surface area.  We assume that each droplet contains a representative surface area distribution as discussed above. This process results in two distributions, one for $A$ and one for $\mu$, with these distributions we can calculate the resultant distribution of $n_s(T)$ values, and from that distribution we obtain the 95% confidence interval.

## 5 Results and discussion

### 5.1 Ice nucleation efficiencies of plagioclase and alkali feldspars

Droplet fraction frozen from µl-NIPI for the 15 feldspar samples are shown in Figure 2. The values of $n_s(T)$ derived from these experiments are shown in Figure 3 along with the $n_s(T)$ parameterisation from Atkinson et al. (2013) for BCS 376 microcline. The various groups of feldspars are indicated by colour which corresponds to the regions of the phase diagram in Figure 1. We define potassium (K-) feldspars (red) as those rich in K including microcline, orthoclase and sanidine; the Na end-member is albite (green); and plagioclase series feldspars (blue) are a solid solution between albite and the calcium end-member, anorthite.

Out of the six K-feldspars studied, five fall on or near the line defined by Atkinson et al. (2013). These include three microcline samples and one sanidine sample, which have different crystal structures. Sanidine has disordered Al atoms, microcline has ordered Al atoms and orthoclase has intermediate order; these differences result in differences in symmetry and hence space group (see Tables 1 and 2). The freezing results indicate that Al disordering does not play an important role in nucleation for the analysed weight concentration range. However, one K-feldspar sample, TUD#3 microcline, was substantially more active.  This indicates that crystal structure and composition are not the only factors dictating the ice nucleating ability of K-feldspars.

All plagioclase feldspars tested were less active ice nucleators than the K-feldspars which were tested. There was relatively little variation in the ice nucleation activities of the plagioclase solid solution series characterised by Carpenter (1986) and Carpenter et al. (1985). For instance, of those feldspars that possess the plagioclase structure, greater sodium content does not systematically increase effectiveness of ice nucleation. Overall, the results

for plagioclase feldspars indicate that they have an ice nucleating ability much smaller than that of the K-feldspars.

It is also interesting to note that the ANC 68 synthetic anorthite had different nucleating properties to the anorthite glass from which it was crystallised (and had the same composition). The ANC 68 synthetic anorthite sample has a much more shallow $n_s(T)$ curve than the glass. This is noteworthy, because the composition of these two materials is identical, but the phase of the material is different. It demonstrates that crystallinity is not required to cause nucleation, but the presence of crystallinity can provide rare active sites which can trigger nucleation at much higher temperatures. In a future study it would be interesting to attempt to probe the nature of these active sites.

We tested three predominantly Na-feldspars (albites). Amelia albite was found to be highly active, approaching that of TUD#3 microcline. The others, BCS 375 albite, and TUD#2 albite were less active, intermediate between the K-feldspars and plagioclase feldspars.

To ensure that the high activity of Amelia albite and microcline TUD#3 was not caused by contamination from biological INPs the samples were heated to 100°C in Milli-Q water for 15 minutes. This treatment will disrupt any protein based nucleators present (O′Sullivan et al., 2015). No significant reduction in freezing temperatures (beyond what would be expected from the activity decay described in Sect. 5.2) was observed suggesting that the highly active INPs present are associated with the feldspars rather than biological protein contamination. Certain biological nucleators have been observed to retain their ice nucleating activity despite heat treatment of this type (Pummer et al., 2012;O'Sullivan et al., 2014;Tobo et al., 2014) however, to the best of our knowledge, no biological species has been observed to nucleate ice at such warm temperatures after heat treatment. This behaviour does not seem consistent with biological nucleators, unless the biological entity is within the Amelia albite particles and is somehow dispersed through the particle population during grinding. While we cannot exclude the possibility that some unknown biological species is present on microcline TUD#3 and Amelia albite it seems more likely that the minerals themselves are responsible for the observed ice nucleation activity. Additionally, it is known that certain organic molecules can nucleate ice efficiently (Fukuta, 1966). It is not possible to exclude the possibility of the presence of these or other, unknown, heat resistant contaminants that nucleate ice very efficiently.

It has been noted by Vali (2014) that there is an indication that nucleators which are more active at higher temperatures tend to have steeper slopes of ln $J$ (nucleation rate). We have observed this trend here in the data shown in Figure 3 ($n_s(T)$ is proportional to $J$ for a single component). The slopes of experiments where freezing occurred at lower temperatures (plagioclases) generally being flatter than those where freezing took place at higher temperatures (alkali feldspars). Vali (2014) suggests that this maybe the result of different observational methods. In this study we have used a single method for all experiments so the trend is unlikely to be due to an instrument artefact. The implication is that active sites with lower activity tend to be more diverse in nature. This may indicate that there are fewer possible ways to compose an active site that is efficient at nucleating ice and that there will be less variation in these sites as a result. The active sites of lower activity may take a greater range of forms and so encompass a greater diversity of freezing temperatures. The lower diversity in the sites active at higher temperatures may explain the steep slopes in $n_s$ seen, however it should be noted that classical nucleation theory also predicts steeper slopes at higher temperatures assuming a single contact angle.

To summarise, plagioclase feldspars tend to have relatively poor ice nucleating abilities, all K-feldspars we tested are relatively good at nucleating ice and the albites are variable in their nucleating activity. Out of the six K-feldspars tested, five have very similar activities and are well approximated by the parameterisation of Atkinson et al. (2013) in the temperature-$n_s$ regime we investigated here. However, we have identified two alkali feldspar samples, one K-feldspar and one albite, which are much more active than the others indicating that a factor or factors other than the polymorph or composition determines the efficiency of alkali feldspars as ice nucleators.

## 5.2 The stability of active sites

It was observed that the ice nucleation activity of ground Amelia albite and ground TUD #3 microcline declined over the course of ~30 minutes. Only the initial run is shown in Figure 3 where the feldspar had spent only about 10 minutes in suspension. This decay in activity over the course of ~30 mins was not seen in the other feldspars. To investigate this effect samples of BCS 376 microcline, Amelia albite and TUD #3 microcline were left in water within a sealed vial and tested at intervals over the course of 16 months, with a focus on the first 11 days. TUD #3 microcline and Amelia albite were chosen for this experiment as they contained highly active sites, represented two different types of feldspar and were the only

feldspars observed to exhibit this rapid decay in activity. BCS 376 microcline was also included in this activity decay experiment as it had provided consistent data over repeated runs and served as a standard in the Atkinson et al. (2013) paper which could therefore be tested. The results of these experiments are shown in Figure 4. The median freezing temperature of the Amelia albite sample was most sensitive to time spent in water, decreasing by 8 $^o$C in 11 days and by 16 $^o$C in 16 months. The TUD#3 microcline sample decreased by about 2 $^o$C in 16 months, but the freezing temperatures of the BCS 376 did not change significantly over 16 months (within the temperature uncertainty of $\pm$0.4°C). Clearly, the stability of the active sites responsible for ice nucleation in these samples is highly variable.

Amelia albite is a particularly interesting case, where the highly active sites are also highly unstable. For Amelia albite we observed that the ice nucleation ability of the powder directly as supplied (the sample had been ground many years prior to experiments) was much lower than the freshly ground sample. The $n_s$ values for the 'as-supplied' Amelia albite are shown in Figure 4. This suggests that the active sites on Amelia albite are unstable and in general are sensitive to the history of the sample. We note that from previous work that BCS 376 feldspar ground to varied extents nucleates ice similarly (Whale et al., 2015) and we have not observed a decay of active sites of the BCS 376 microcline sample when stored in a dry vial over the course of two years. It is also worth noting that freshly ground BCS 376 microcline did not nucleate ice as efficiently as Amelia albite or TUD#3 microcline. These results indicate that BCS 376 microcline contains very active sites, but that these sites are much more stable than those found in Amelia albite. This result is in agreement with the observation that albite weathers faster than microcline in soils as Na$^+$ is more readily substituted for hydrogen than K$^+$ (Busenberg and Clemency, 1976;Blum, 1994).

Zolles at al. (2015) have suggested that grinding can lead to active sites being revealed, or the enhancement of existing active sites. It was shown in Whale et al. (2015) that differently ground samples of BCS 376 microcline nucleate ice similarly. In contrast Hiranuma et al. (2014) show that ground hematite nucleates ice more efficiently (normalised to surface area) than cubic hematite. The evidence suggests that the ice nucleating efficiencies of different materials respond differently to grinding processes. Indeed, it is evident from this study that highly active sites in Amelia albite are generated by grinding but lose activity when exposed to liquid water, and probably lose activity during exposure to (presumably wet) air, returning to an activity level comparable to that of the plagioclase feldspars. TUD#3 microcline also possesses a highly active site type sensitive to water exposure but falls back to a level of

activity higher than the other K-feldspars we have tested. This second, less active site type is shown to be stable in water over the course of 16 months. TUD#3 must possess populations of both more active, unstable sites and less active (although still relatively active compared to the sites on other K-feldspars) stable sites. Amelia albite possesses only unstable sites and much less active sites similar to those found on the plagioclase feldspars we have tested.

These results indicate something of the nature of the active sites on feldspars. Throughout this paper we refer to nucleation occurring on active sites, or specific sites, on the surface of feldspar. It is thought that nucleation by most ice active minerals is consistent with nucleation on active sites with a broad spectrum of activities (Marcolli et al., 2007;Lüönd et al., 2010;Niedermeier et al., 2010;Augustin-Bauditz et al., 2014;Herbert et al., 2014;Vali, 2014;Wex et al., 2014;Wheeler et al., 2015;Hiranuma et al., 2015;Niedermeier et al., 2015;Hartmann et al., 2016). However, the nature of these active sites is not known.  It is postulated that active sites are related to defects in the structure and therefore that each site has a characteristic nucleation ability, producing a spectrum of active sites. Defects are inherently less stable than the bulk of the crystal and we might expect these sites to be affected by dissolution processes, or otherwise altered, in preference to the bulk of the crystal (Parsons et al., 2015). The fact that we observe ice nucleation by populations of active sites with different stabilities in water implies that these sites have different physical or chemical characteristics. Furthermore, the fact that some populations of active sites are sensitive to exposure to water suggests that the history of particles can be critical in determining the ice nucleating ability of mineral dusts. This raises the question of whether differences in ice nucleation efficiency observed by different instruments (Emersic et al., 2015;Hiranuma et al., 2015), could be related to the different conditions particles experience prior to nucleation.

## 5.3 Comparison to literature data

We have compared the $n_s(T)$ values for various feldspars from a range of literature sources with data from this study in Figure 5. Inspection of this plot confirms that K-feldspars nucleate ice more efficiently than the plagioclase feldspars.  Also, with the exception of the hyper-active Amelia albite sample, the K-feldspars are more active than the albites.

Results for BCS 376 microcline have been reported in several papers (Atkinson et al., 2013;O'Sullivan et al., 2014;Whale et al., 2015;Emersic et al., 2015). There is a discrepancy between the cloud chamber data from Emersic et al. (2015) and the picolitre droplet cold stage experiments at around -18°C,  whereas the data at about -25°C are in agreement.

Emersic et al. (2015) attribute this discrepancy to aggregation of feldspar particles in
microlitre scale droplet freezing experiments reducing the surface area of feldspar exposed to
water leading to a lower $n_s(T)$ value. It is unlikely that this effect can account for the
discrepancy because in the temperature range of the Emersic et al. (2015) data the
comparison is being made to results from picolitre droplet freezing experiments which
Emersic et al. (2015) argue should not be affected by aggregation because there are not
enough particles present in each droplet to result in significant aggregation. Atkinson et al.
(2013) estimated that on average even the largest droplets only contained a few 10s of
particles. We also note that our microscope images of droplets show many individual
particles moving independently around in the picolitre droplets in those experiments,
indicating that the feldspar grains do not aggregate substantially. Hence, the discrepancy
between the data of Emersic et al. (2015) and Atkinson et al. (2013) at around -18°C cannot
be accounted for by aggregation.  Furthermore, Atkinson et al. (2013) report that the surface
area determined from the laser diffraction size distribution of BCS 376 microcline in
suspension is 3.5 times smaller than that derived by the gas adsorption measurements (see
supplementary Figure 5 in Atkinson et al. (2013) and the corresponding discussion).  This
difference in surface area can be accounted for by the fact that feldspar grains are not smooth
spheres, as assumed in the analysis of the laser diffraction data. Feldspar grains are well-
known to be rough and aspherical (Hodson et al., 1997). Atkinson et al. (2013) also note that
the laser diffraction technique lacks sensitivity to the smallest particles in the distribution
which will also lead to an underestimate in surface area. Nevertheless, the data presented by
Atkinson et al. (2013) suggests that aggregation of feldspar particles leading to reduced
surface area is at most a minor effect.  As such the discrepancy between different instruments
remains unexplained and more work is needed on this topic.
Ice nucleation by single size-selected particles of TUD#1 microcline has been investigated by
Niedermeier et al. (2015) at temperatures below -23°C. We found that TUD#1 microcline
was in good agreement with the K-feldspar parameterisation from Atkinson et al. (2013)
between about -6 and -11°C. Between -23 and -25°C, the $n_s(T)$ values produced by
Niedermeier et al. (2015) are similar (lower by a factor of roughly 4) to that of the Atkinson
et al. (2013) parameterisation, despite the different sample types. Niedermeier et al. (2015)
used the Leipzig Aerosol Cloud Interaction Simulator (LACIS), in which they size selected
particles, activated them to cloud droplets and then quantified the probability of freezing at a
particular temperature. It is interesting that the Niedermeier et al. (2015) $n_s(T)$ values curve
off at lower temperatures to a limiting value which they term $n_s^*$, indicating that nucleation
by K-feldspars may hit a maximum value and emphasises why we need to be cautious in
extrapolating $n_s(T)$ parameterisations beyond the range of experimental data.
The data for a microcline, a plagioclase (andesine) and albite from Zolles et al. (2015) is
consistent with our finding that plagioclase feldspars are less effective nucleators than K-
feldspars. It is also consistent with Atkinson et al. (2013) who found that albite is less
efficient at nucleating ice than microcline. However, the data for K-feldspar from Zolles et al.
(2015) sits below the line from Atkinson et al. (2013) for BCS 376 microcline and are lower
than the points from Niedermeier et al. (2015) for TUD#1 microcline. Their measurements
involved making up suspensions (2-5wt%)  and then creating a water-in-oil emulsion where
droplets were between 10-40 µm.  They quote their particle sizes as being between 1-10 µm
for the feldspars. Atkinson et al. (2013) worked with 0.8 wt% suspensions, with droplets of 9
to 19 µm where the mode particle size was ~700 nm. Hence, Zolles et al. (2015) worked with
more concentrated suspensions and larger particles than used by Atkinson et al. (2013). In
principle, $n_s$ should be independent of droplet volume and particle concentration, but
differences between instruments and methods have been reported (Hiranuma et al., 2015).
Additionally, Zolles et al. (2015) estimated the surface area of their feldspar particles using a
combination of SEM images and the BET surface area of quartz. This leads to an unspecified
uncertainty in their $n_s$ values. However, it is not possible to determine whether the observed
difference in $n_s$ is due to differences in the sample or the techniques used, but may mean that
certain K-feldspars nucleate ice less well than those defined by the Atkinson et al. (2013) line
in this temperature regime. This would be a very interesting result as it may provide a point
of difference that could provide insight into why K-feldspars nucleate ice efficiently. There
has been relatively little work on what makes feldspar a good nucleator of ice. Zolles et al.
(2015) suggest that only K-feldspars will nucleate ice well on the basis that $Ca^{2+}$ and $Na^+$ are
chaotropic (structure breaking in water) while $K^+$ is kosmotropic (structure making in water).
We have only observed one feldspar that contains little $K^+$ but nucleates ice relatively
efficiently, Amelia albite. This feldspar loses its activity quickly in water and eventually
becomes more comparable to the plagioclase feldspars. It may be that the strong nucleation
observed is associated with the small amount of $K^+$ it contains and that once this dissolves
away the feldspar behaves like a plagioclase.
Augustin-Bauditz et al. (2014) tentatively concluded that microcline may nucleate ice more
efficiently than orthoclase at $n_s(T)$ values above about $10^6$ cm$^{-2}$ and at temperatures below -

23°C, the conditions where they performed their measurements.  They arrived at this conclusion by noting that NX-illite and Arizona test dust both contain orthoclase (8 and 20%, respectively), but the $n_s(T)$ values they report for these materials are more than one order less than microcline.

Within the surface area regime examined in this study we have observed some variability amongst the K-feldspars (see Figure 2), but no difference between sanidine and four out of five microclines which fall around the line defined by Atkinson et al. (2013). As discussed above, the Al in sanidine is the least ordered, with microcline the most ordered and orthoclase at an intermediate order, hence we observe no clear dependency on the ordering of Al in K-feldspars. Further investigations of the ice nucleating ability of the various K-feldspar phases at low temperature would be valuable. We could not do this in the present study with the samples used here because we did not have sufficient quantities of the samples.

## 6        Conclusions

In this study we have analysed the ice nucleating ability of 15 characterised feldspar samples. These minerals include plagioclase feldspars (in the solid solution series between Ca and Na end-members), the K-feldspars (sanidine and microcline) and albite (the Na end-member). The results indicate that the alkali feldspars, including albite and K-feldspars, tend to nucleate ice more efficiently than plagioclase feldspars. The plagioclase feldspars nucleate ice at the lowest temperatures with no obvious dependence on the Ca-Na ratio. The albites have a wide variety of nucleating abilities, with one sample nucleating ice much more efficiently than the microcline sample Atkinson et al. (2013) studied. This hyper-active albite lost its activity over time while suspended in water. Five out of six of the K-feldspar samples we studied nucleated ice with a similar efficiency to the BCS 376 microcline studied by Atkinson et al. (2013). A single K-feldspar we studied had a very high activity, nucleating ice as warm as -2°C in our microliter droplet assay.  The striking activity of this hyperactive microcline decayed with time spent in water, but not to the same extent as the hyperactive albite sample. While the hyperactive sites are sensitive, to varying degrees, to time spent in water, the activity of the BCS 376 microcline sample used by Atkinson et al. (Atkinson et al., 2013) did not change significantly. We have not excluded the possibility that other entities on the surfaces of the feldspar may be responsible for the ice nucleation observed.

In light of these findings, we suggest that there are at least three classes of active site present in the feldspars studied here: *i*) sites of relatively low activity associated with plagioclase feldspars; *ii*) sites which are more active associated with K-feldspars that are stable in water over the course of many months; *iii*) hyper-active sites associated with one albite and one K-feldspar that we studied that loses activity when exposed to water. It is possible that the sites of type *i* are present on the typical K-feldspars, but we do not observe them because ice nucleates on more active sites. Whether these different sites are all related to similar features on the surfaces or if they are each related to different types of features is not known. Nevertheless, it appears that feldspars are characterised by a range of active site types with varying stability and activity.

The specific details of these active sites continue to elude us, although it appears that they are only present in alkali feldspars and in particular, the K-feldspars. Unlike the plagioclase feldspars which form a solid solution, the Na and K feldspars in alkali feldspars are often exsolved, possessing intergrowths of the Na and K feldspars referred to as microtexture (Parsons et al., 2015). It is possible that the boundaries between the two phases in the intergrowth provide sites for nucleation that are not present in plagioclase feldspars. If the high energy defects along exsolution boundaries are responsible for higher ice nucleation activity of K-feldspars then this may offer an insight into acid passivation of ice nucleating ability observed in laboratory studies (Wex et al., 2014;Augustin-Bauditz et al., 2014). Berner and Holdren (1979) suggest that the acid mediated weathering of feldspar occurs in multiple stages and suggest dissolution of feldspars is concentrated at high surface energy sites such as dislocations and crystal defects, sites which may be related to ice nucleation. More work is needed to explore the significance of exsolution, microtexture and the impact of weathering on feldspars with respect to ice nucleation activity.

In a previous study Atkinson et al. (2013) used an $n_s(T)$ parameterisation of a single K-feldspar (BCS 376 microcline) to approximate the ice nucleating properties of desert dust in a global aerosol model. Given that five out of six of the K-feldspars we studied here have very similar ice nucleating abilities, this approximation seems reasonable. However, we have identified two hyper-active feldspars and do not know how representative these samples are of natural feldspars in dust emission regions. We also note that the active sites on these feldspars are less stable than those of BCS 376 microcline. Nevertheless, there is the possibility that the parameterisation used by Atkinson et al. (2013) underestimates the contribution of feldspars at higher temperatures above about -15°C.

In the longer term it may be possible to identify what it is that leads to the variation in ice nucleation activity between the different feldspar classes. In particular, the nature of the active sites in the hyper-active feldspars and the reason plagioclase is so much poorer at nucleating ice are subjects of interest. The instability of the active sites in the hyperactive feldspars may be related to dissolution of feldspar in water and investigation of this process may allow progress towards understanding of nucleation by feldspars. The results presented here are empirical in nature and do not provide a thorough underpinning understanding of the nature of the active sites. Nevertheless, the fact that the feldspar group of minerals have vastly different ice nucleating properties despite possessing very similar crystal structures may provide us with a means of gaining a fundamental insight to heterogeneous ice nucleation.

**Acknowledgments**

We would like to acknowledge Theodore Wilson and Alexei Kiselev for helpful discussions and John Morris for introducing TFW and MAC. We are grateful to Alexei Kiselev and Martin Ebert for providing the TUD samples. Alex Harrison thanks the School of Earth and Environment for an Undergraduate Research Scholarship which allowed him to make many of the measurements presented in this paper. We would like to thank the National Environmental Research Council, (NERC, NE/I013466/1; NE/I020059/1; NE/K004417/1; NE/I019057/1; NE/M010473/1) the European Research Council (ERC, 240449 ICE; 632272 IceControl; 648661 MarineIce), and the Engineering and Physical Sciences Research Council (EPSRC, EP/M003027/1) for funding.

Alpert, P. A., and Knopf, D. A.: Analysis of isothermal and cooling-rate-dependent immersion freezing by a unifying stochastic ice nucleation model, Atmos. Chem. Phys., 16, 2083-2107, 10.5194/acp-16-2083-2016, 2016.

Atkinson, J. D., Murray, B. J., Woodhouse, M. T., Whale, T. F., Baustian, K. J., Carslaw, K. S., Dobbie, S., O'Sullivan, D., and Malkin, T. L.: The importance of feldspar for ice nucleation by mineral dust in mixed-phase clouds, Nature, 498, 355-358, 10.1038/nature12278, 2013.

Augustin-Bauditz, S., Wex, H., Kanter, S., Ebert, M., Niedermeier, D., Stolz, F., Prager, A., and
Stratmann, F.: The immersion mode ice nucleation behavior of mineral dusts: A comparison of
different pure and surface modified dusts, Geophys. Res. Lett., 41, 7375-7382,
10.1002/2014gl061317, 2014.
Berner, R. A., and Holdren, G. R.: Mechanism of feldspar weathering—ii. Observations of feldspars
from soils, Geochim. Cosmochim. Acta, 43, 1173-1186, http://dx.doi.org/10.1016/0016-
8    7037(79)90110-8, 1979.

Blum, A. E.: Feldspars in weathering, in: Feldspars and their reactions, Springer, 595-630, 1994.
Busenberg, E., and Clemency, C. V.: The dissolution kinetics of feldspars at 25 c and 1 atm co2 partial
pressure, Geochim. Cosmochim. Acta, 40, 41-49, 1976.
Carpenter, M.: Experimental delineation of the "e" $\rightleftharpoons$ i\bar 1 and "e" $\rightleftharpoons$ c\bar 1 transformations in
intermediate plagioclase feldspars, Phys Chem Minerals, 13, 119-139, 10.1007/bf00311902, 1986.
Carpenter, M. A., McConnell, J. D. C., and Navrotsky, A.: Enthalpies of ordering in the plagioclase
feldspar solid solution, Geochim. Cosmochim. Acta, 49, 947-966, http://dx.doi.org/10.1016/0016-
20   7037(85)90310-2, 1985.

Carpenter, M. A.: Mechanisms and kinetics of al-si ordering in anorthite; i, incommensurate
structure and domain coarsening, American Mineralogist, 76, 1110-1119, 1991.
Connolly, P. J., Möhler, O., Field, P. R., Saathoff, H., Burgess, R., Choularton, T., and Gallagher, M.:
Studies of heterogeneous freezing by three different desert dust samples, Atmos. Chem. Phys., 9,
2805-2824, 10.5194/acp-9-2805-2009, 2009.
Cox, S. J., Kathmann, S. M., Purton, J. A., Gillan, M. J., and Michaelides, A.: Non-hexagonal ice at
hexagonal surfaces: The role of lattice mismatch, Phys. Chem. Chem. Phys., 14, 7944-7949,
10.1039/c2cp23438f, 2012.
Cox, S. J., Kathmann, S. M., Slater, B., and Michaelides, A.: Molecular simulations of heterogeneous
ice nucleation. I. Controlling ice nucleation through surface hydrophilicity, The Journal of Chemical
Physics, 142, 184704, doi:http://dx.doi.org/10.1063/1.4919714, 2015a.

Cox, S. J., Kathmann, S. M., Slater, B., and Michaelides, A.: Molecular simulations of heterogeneous
ice nucleation. Ii. Peeling back the layers, The Journal of Chemical Physics, 142, 184705,
doi:http://dx.doi.org/10.1063/1.4919715, 2015b.

Deer, W. A., Howie, R. A., and Zussman, J.: An introduction to the rock forming minerals, 2nd ed.,
Addison Wesley Longman, Harlow, UK, 1992.

DeMott, P. J., Sassen, K., Poellot, M. R., Baumgardner, D., Rogers, D. C., Brooks, S. D., Prenni, A. J., and Kreidenweis, S. M.: African dust aerosols as atmospheric ice nuclei, Geophys. Res. Lett., 30, 1732, 10.1029/2003GL017410, 2003.

Emersic, C., Connolly, P. J., Boult, S., Campana, M., and Li, Z.: Investigating the discrepancy between wet-suspension- and dry-dispersion-derived ice nucleation efficiency of mineral particles, Atmos. Chem. Phys., 15, 11311-11326, 10.5194/acp-15-11311-2015, 2015.

Fitzner, M., Sosso, G. C., Cox, S. J., and Michaelides, A.: The many faces of heterogeneous ice nucleation: Interplay between surface morphology and hydrophobicity, J. Am. Chem. Soc., 137, 13658-13669, 10.1021/jacs.5b08748, 2015.

Freedman, M. A.: Potential sites for ice nucleation on aluminosilicate clay minerals and related materials, The Journal of Physical Chemistry Letters, 2015.

Fukuta, N.: Experimental studies of organic ice nuclei, J. Atmos. Sci., 23, 191-196, doi:10.1175/1520-0469(1966)023<0191:ESOOIN>2.0.CO;2, 1966.

Ginoux, P., Prospero, J. M., Gill, T. E., Hsu, N. C., and Zhao, M.: Global-scale attribution of anthropogenic and natural dust sources and their emission rates based on modis deep blue aerosol products, Rev. Geophys., 50, RG3005, 10.1029/2012rg000388, 2012.

Glaccum, R. A., and Prospero, J. M.: Saharan aerosols over the tropical north-atlantic - mineralogy, Mar. Geol., 37, 295-321, 10.1016/0025-3227(80)90107-3, 1980.

Hartmann, S., Wex, H., Clauss, T., Augustin-Bauditz, S., Niedermeier, D., Rösch, M., and Stratmann, F.: Immersion freezing of kaolinite: Scaling with particle surface area, J. Atmos. Sci., 73, 263-278, doi:10.1175/JAS-D-15-0057.1, 2016.

Herbert, R. J., Murray, B. J., Whale, T. F., Dobbie, S. J., and Atkinson, J. D.: Representing time-dependent freezing behaviour in immersion mode ice nucleation, Atmos. Chem. Phys., 14, 8501-8520, 10.5194/acp-14-8501-2014, 2014.

Herbert, R. J., Murray, B. J., Dobbie, S. J., and Koop, T.: Sensitivity of liquid clouds to homogenous freezing parameterizations, Geophys. Res. Lett., 42, 1599-1605, 10.1002/2014gl062729, 2015.

Hiranuma, N., Hoffmann, N., Kiselev, A., Dreyer, A., Zhang, K., Kulkarni, G., Koop, T., and Möhler, O.: Influence of surface morphology on the immersion mode ice nucleation efficiency of hematite particles, Atmos. Chem. Phys., 14, 2315-2324, 10.5194/acp-14-2315-2014, 2014.

Hiranuma, N., Augustin-Bauditz, S., Bingemer, H., Budke, C., Curtius, J., Danielczok, A., Diehl, K., Dreischmeier, K., Ebert, M., Frank, F., Hoffmann, N., Kandler, K., Kiselev, A., Koop, T., Leisner, T., Möhler, O., Nillius, B., Peckhaus, A., Rose, D., Weinbruch, S., Wex, H., Boose, Y., DeMott, P. J., Hader,

J. D., Hill, T. C. J., Kanji, Z. A., Kulkarni, G., Levin, E. J. T., McCluskey, C. S., Murakami, M., Murray, B. J., Niedermeier, D., Petters, M. D., O'Sullivan, D., Saito, A., Schill, G. P., Tajiri, T., Tolbert, M. A., Welti, A., Whale, T. F., Wright, T. P., and Yamashita, K.: A comprehensive laboratory study on the immersion freezing behavior of illite nx particles: A comparison of 17 ice nucleation measurement techniques, Atmos. Chem. Phys., 15, 2489-2518, 10.5194/acp-15-2489-2015, 2015.

Hodson, M. E., Lee, M. R., and Parsons, I.: Origins of the surface roughness of unweathered alkali feldspar grains, Geochim. Cosmochim. Acta, 61, 3885-3896, 10.1016/s0016-7037(97)00197-x, 1997.

Hoose, C., Lohmann, U., Erdin, R., and Tegen, I.: The global influence of dust mineralogical composition on heterogeneous ice nucleation in mixed-phase clouds, Environ. Res. Lett., 3, 10.1088/1748-9326/3/2/025003, 2008.

Hoose, C., Kristjánsson, J. E., Chen, J.-P., and Hazra, A.: A classical-theory-based parameterization of heterogeneous ice nucleation by mineral dust, soot, and biological particles in a global climate model, J. Atmos. Sci., 67, 2483-2503, 10.1175/2010jas3425.1, 2010.

Hoose, C., and Möhler, O.: Heterogeneous ice nucleation on atmospheric aerosols: A review of results from laboratory experiments, Atmos. Chem. Phys., 12, 9817-9854, 10.5194/acp-12-9817-2012, 2012.

Hu, X. L., and Michaelides, A.: Ice formation on kaolinite: Lattice match or amphoterism?, Surface Science, 601, 5378-5381, 10.1016/j.susc.2007.09.012, 2007.

Kandler, K., Benker, N., Bundke, U., Cuevas, E., Ebert, M., Knippertz, P., Rodríguez, S., Schütz, L., and Weinbruch, S.: Chemical composition and complex refractive index of saharan mineral dust at izana, tenerife (spain) derived by electron microscopy, Atmos. Environ., 41, 8058-8074, 2007.

Kandler, K., Schütz, L., Deutscher, C., Ebert, M., Hofmann, H., Jäckel, S., Jaenicke, R., Knippertz, P., Lieke, K., Massling, A., Petzold, A., Schladitz, A., Weinzierl, B., Wiedensohler, A., Zorn, S., and Weinbruch, S.: Size distribution, mass concentration, chemical and mineralogical composition and derived optical parameters of the boundary layer aerosol at tinfou, morocco, during samum 2006, Tellus, 61B, 32-50, 10.1111/j.1600-0889.2008.00385.x, 2009.

Kandler, K., SchÜTz, L., JÄCkel, S., Lieke, K., Emmel, C., MÜLler-Ebert, D., Ebert, M., Scheuvens, D., Schladitz, A., ŠEgviĆ, B., Wiedensohler, A., and Weinbruch, S.: Ground-based off-line aerosol measurements at praia, cape verde, during the saharan mineral dust experiment: Microphysical properties and mineralogy, Tellus, 63B, 459-474, 10.1111/j.1600-0889.2011.00546.x, 2011.

Lüönd, F., Stetzer, O., Welti, A., and Lohmann, U.: Experimental study on the ice nucleation ability of size-selected kaolinite particles in the immersion mode, J. Geophys. Res., 115, D14201, 10.1029/2009jd012959, 2010.

Lupi, L., Hudait, A., and Molinero, V.: Heterogeneous nucleation of ice on carbon surfaces, J. Am.
Chem. Soc., 136, 3156-3164, 10.1021/ja411507a, 2014.
Lupi, L., and Molinero, V.: Does hydrophilicity of carbon particles improve their ice nucleation
ability?, J. Phys. Chem. A, 118, 7330-7337, 10.1021/jp4118375, 2014.
Marcolli, C., Gedamke, S., Peter, T., and Zobrist, B.: Efficiency of immersion mode ice nucleation on
surrogates of mineral dust, Atmos. Chem. Phys., 7, 5081-5091, 10.5194/acp-7-5081-2007, 2007.
Murray, B. J., Broadley, S. L., Wilson, T. W., Atkinson, J. D., and Wills, R. H.: Heterogeneous freezing
of water droplets containing kaolinite particles, Atmos. Chem. Phys., 11, 4191-4207, 10.5194/acp-
12  11-4191-2011, 2011.

Murray, B. J., O'Sullivan, D., Atkinson, J. D., and Webb, M. E.: Ice nucleation by particles immersed in
supercooled cloud droplets, Chem. Soc. Rev., 41, 6519-6554, 10.1039/C2CS35200A, 2012.
Niedermeier, D., Hartmann, S., Shaw, R. A., Covert, D., Mentel, T. F., Schneider, J., Poulain, L., Reitz,
P., Spindler, C., Clauss, T., Kiselev, A., Hallbauer, E., Wex, H., Mildenberger, K., and Stratmann, F.:
Heterogeneous freezing of droplets with immersed mineral dust particles - measurements and
parameterization, Atmos. Chem. Phys., 10, 3601-3614, 10.5194/acp-10-3601-2010, 2010.
Niedermeier, D., Augustin-Bauditz, S., Hartmann, S., Wex, H., Ignatius, K., and Stratmann, F.: Can we
define an asymptotic value for the ice active surface site density for heterogeneous ice nucleation?,
Journal of Geophysical Research: Atmospheres, 120, 5036-5046, 10.1002/2014jd022814, 2015.
Niemand, M., Möhler, O., Vogel, B., Vogel, H., Hoose, C., Connolly, P., Klein, H., Bingemer, H.,
DeMott, P. J., Skrotzki, J., and Leisner, T.: A particle-surface-area-based parameterization of
immersion freezing on desert dust particles, J. Atmos. Sci., 69, 10.1175/jas-d-11-0249.1, 2012.
O'Sullivan, D., Murray, B. J., Malkin, T. L., Whale, T. F., Umo, N. S., Atkinson, J. D., Price, H. C.,
Baustian, K. J., Browse, J., and Webb, M. E.: Ice nucleation by fertile soil dusts: Relative importance
of mineral and biogenic components, Atmos. Chem. Phys., 14, 1853-1867, 10.5194/acp-14-1853-
33  2014, 2014.

O′ Sullivan, D., Murray, B. J., Ross, J. F., Whale, T. F., Price, H. C., Atkinson, J. D., Umo, N. S., and
Webb, M. E.: The relevance of nanoscale biological fragments for ice nucleation in clouds, Sci. Rep.,
5, 10.1038/srep08082, 2015.

Parsons, I., Fitz Gerald, J. D., and Lee, M. R.: Routine characterization and interpretation of complex
alkali feldspar intergrowths, American Mineralogist, 100, 1277-1303, 10.2138/am-2015-5094, 2015.

Perlwitz, J. P., Pérez García-Pando, C., and Miller, R. L.: Predicting the mineral composition of dust
aerosols – part 1: Representing key processes, Atmos. Chem. Phys., 15, 11593-11627, 10.5194/acp-
3    15-11593-2015, 2015.

Pummer, B. G., Bauer, H., Bernardi, J., Bleicher, S., and Grothe, H.: Suspendable macromolecules are
responsible for ice nucleation activity of birch and conifer pollen, Atmos. Chem. Phys., 12, 2541-
2550, 10.5194/acp-12-2541-2012, 2012.
Reinhardt, A., and Doye, J. P. K.: Effects of surface interactions on heterogeneous ice nucleation for a
monatomic water model, Journal of Chemical Physics, 141, 10.1063/1.4892804, 2014.
Riechers, B., Wittbracht, F., Hütten, A., and Koop, T.: The homogeneous ice nucleation rate of water
droplets produced in a microfluidic device and the role of temperature uncertainty, Phys. Chem.
Chem. Phys., 15, 5873-5887, 10.1039/C3CP42437E, 2013.
Slater, B., Michaelides, A., Salzmann, C. G., and Lohmann, U.: A blue-sky approach to understanding
cloud formation, B. Am. Meteorol. Soc., 10.1175/bams-d-15-00131.1, 2015.
Tobo, Y., DeMott, P. J., Hill, T. C. J., Prenni, A. J., Swoboda-Colberg, N. G., Franc, G. D., and
Kreidenweis, S. M.: Organic matter matters for ice nuclei of agricultural soil origin, Atmos. Chem.
Phys., 14, 8521-8531, 10.5194/acp-14-8521-2014, 2014.
Umo, N. S., Murray, B. J., Baeza-Romero, M. T., Jones, J. M., Lea-Langton, A. R., Malkin, T. L.,
O'Sullivan, D., Neve, L., Plane, J. M. C., and Williams, A.: Ice nucleation by combustion ash particles at
conditions relevant to mixed-phase clouds, Atmos. Chem. Phys., 15, 5195-5210, 10.5194/acp-15-
26   5195-2015, 2015.

Vali, G.: Repeatability and randomness in heterogeneous freezing nucleation, Atmos. Chem. Phys., 8,
5017-5031, 10.5194/acp-8-5017-2008, 2008.
Vali, G.: Interpretation of freezing nucleation experiments: Singular and stochastic; sites and
surfaces, Atmos. Chem. Phys., 14, 5271-5294, 10.5194/acp-14-5271-2014, 2014.

Vali, G., DeMott, P., Möhler, O., and Whale, T.: Ice nucleation terminology, Atmospheric Chemistry
and Physics Discussions, 14, 22155-22162, 2014.

Wenk, H.-R., and Bulakh, A.: Minerals: Their constitution and origin, Cambridge University Press,
38   2004.

Wex, H., DeMott, P. J., Tobo, Y., Hartmann, S., Rösch, M., Clauss, T., Tomsche, L., Niedermeier, D.,
and Stratmann, F.: Kaolinite particles as ice nuclei: Learning from the use of different kaolinite
samples and different coatings, Atmos. Chem. Phys., 14, 5529-5546, 10.5194/acp-14-5529-2014,
43   2014.

Whale, T. F., Murray, B. J., O'Sullivan, D., Wilson, T. W., Umo, N. S., Baustian, K. J., Atkinson, J. D., Workneh, D. A., and Morris, G. J.: A technique for quantifying heterogeneous ice nucleation in microlitre supercooled water droplets, Atmos. Meas. Tech., 8, 2437-2447, 10.5194/amt-8-2437-2015, 2015.

Wheeler, M. J., Mason, R. H., Steunenberg, K., Wagstaff, M., Chou, C., and Bertram, A. K.: Immersion freezing of supermicron mineral dust particles: Freezing results, testing different schemes for describing ice nucleation, and ice nucleation active site densities, The Journal of Physical Chemistry A, 119, 4358-4372, 10.1021/jp507875q, 2015.

Wittke, W., and Sykes, R.: Rock mechanics, Springer Berlin, 1990.

Wright, T. P., and Petters, M. D.: The role of time in heterogeneous freezing nucleation, J. Geophys. Res.-Atmos., 118, 3731-3743, 10.1002/jgrd.50365, 2013.

Wright, T. P., Petters, M. D., Hader, J. D., Morton, T., and Holder, A. L.: Minimal cooling rate dependence of ice nuclei activity in the immersion mode, J. Geophys. Res.-Atmos., 118, 10535-10543, 10.1002/jgrd.50810, 2013.

Yakobi-Hancock, J. D., Ladino, L. A., and Abbatt, J. P. D.: Feldspar minerals as efficient deposition ice nuclei, Atmos. Chem. Phys., 13, 11175-11185, 10.5194/acp-13-11175-2013, 2013.

Zielke, S. A., Bertram, A. K., and Patey, G. N.: Simulations of ice nucleation by kaolinite (001) with rigid and flexible surfaces, The Journal of Physical Chemistry B, 10.1021/acs.jpcb.5b09052, 2015.

Zolles, T., Burkart, J., Häusler, T., Pummer, B., Hitzenberger, R., and Grothe, H.: Identification of ice nucleation active sites on feldspar dust particles, The Journal of Physical Chemistry A, 119, 2692-2700, 10.1021/jp509839x, 2015.

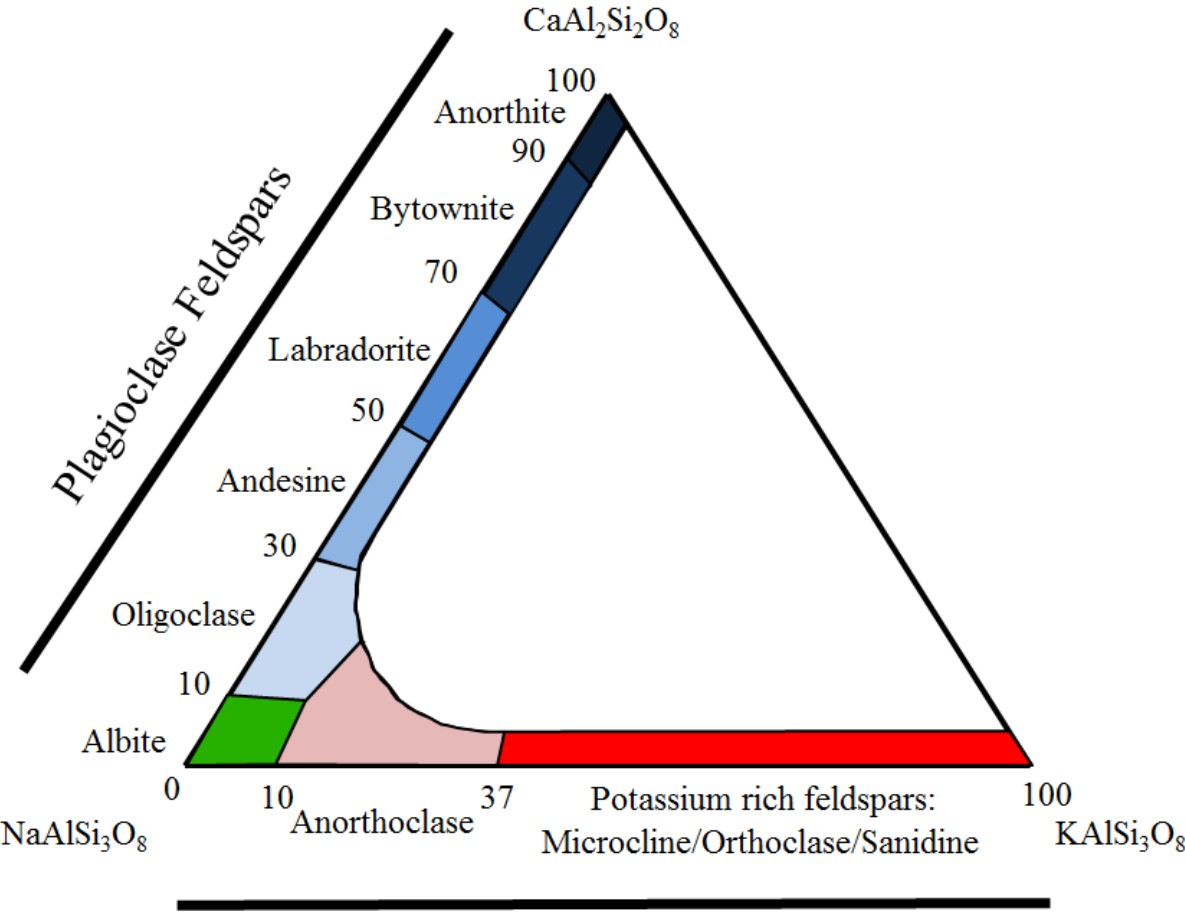

3  **Figure 1**: The ternary composition diagram for the feldspars group based on similar figures
4  in the literature (Wittke and Sykes, 1990;Deer et al., 1992).

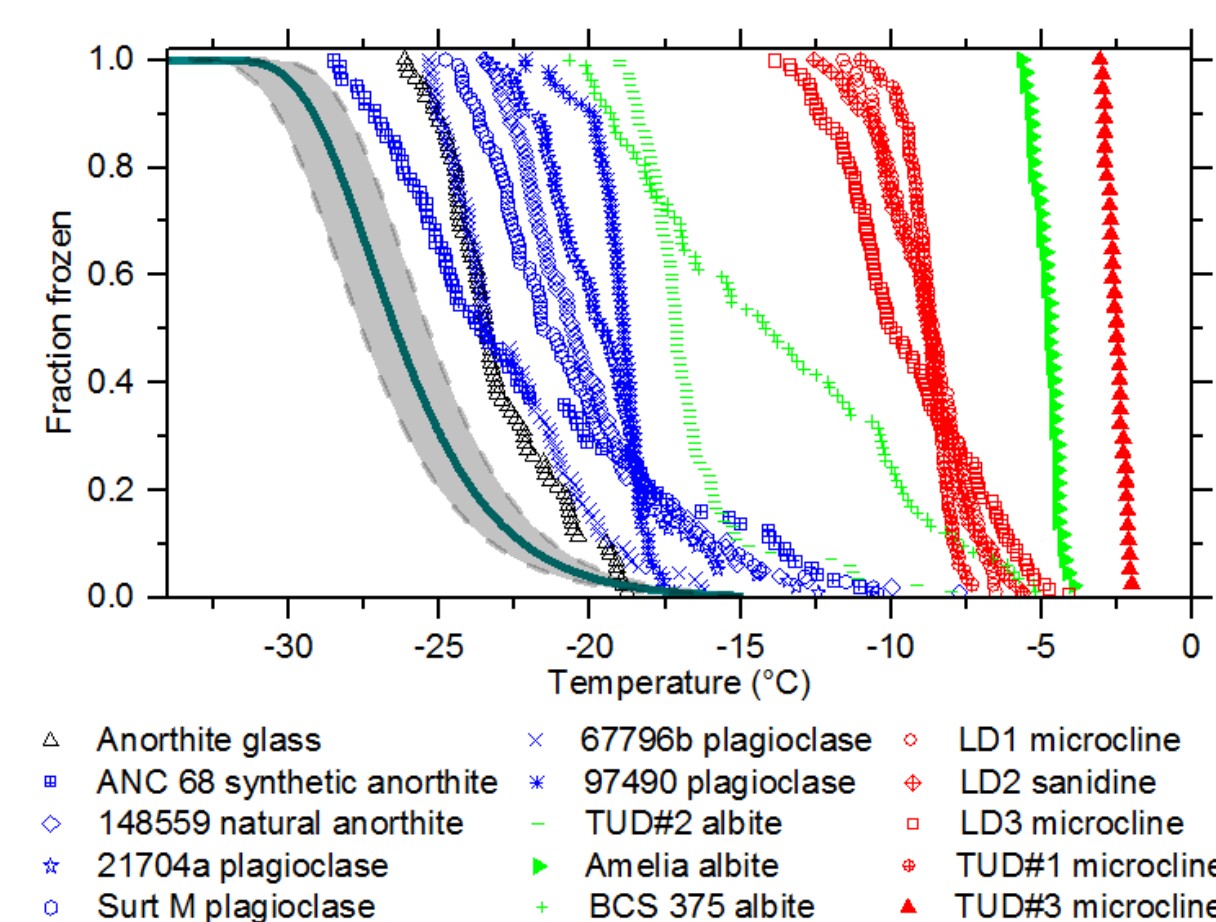

**Figure 2:** Droplet fraction frozen as a function of temperature for 1 wt% suspensions of
ground powders of various feldspar samples. The K-feldspars are coloured red, the
plagioclase feldspars are coloured blue, the albites are coloured green and the feldspar glass
is coloured black.  A fit to the background freezing of pure MilliQ water in the µl-NIPI
instrument used by Umo et al. (2015) is also included. The shaded area around this fit shows
95% confidence intervals for the fit. It can be seen that all the feldspar samples tested
nucleate ice more efficiently than the background freezing of the instrument.

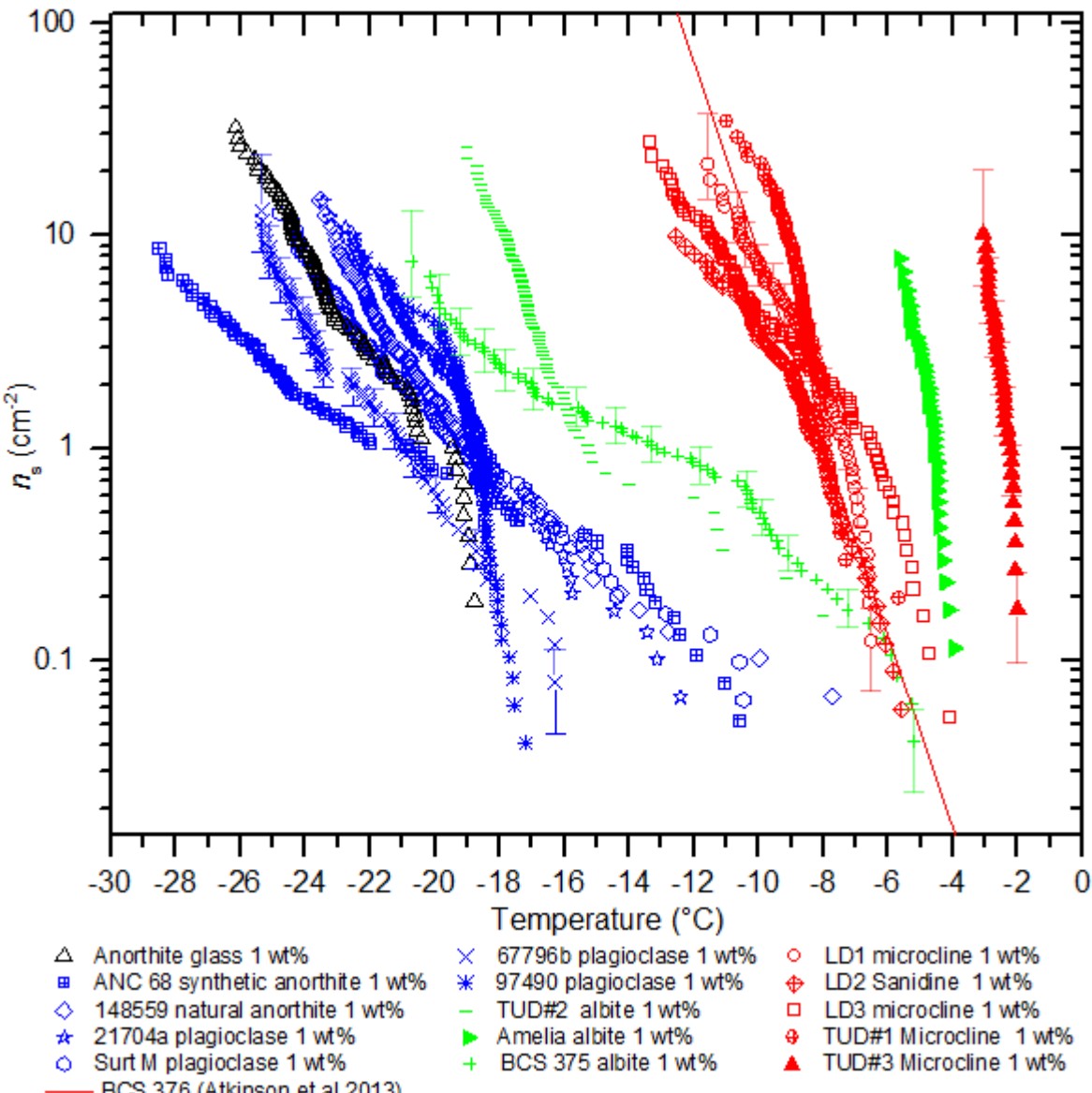

**Figure 3**: Ice nucleation efficiency expressed as $n_s$ $(T)$ for the various feldspars tested in this
study. The K-feldspars are coloured red, the plagioclase feldspars are coloured blue, the
albites are coloured green and the feldspar glass is coloured black. Except for Amelia albite
and TUD#1 microcline all samples were tested twice and the data from the two runs
combined.   Sample information can be found in tables 1 and 2. Temperature uncertainty is
±0.4°C. Y-Error bars calculated using the Poisson Monte Carlo procedure described in Sect.
4. Data points with large uncertainties greater than an order of magnitude have been removed,
these are invariably the first one or two freezing events of a given experiment. For clarity
error bars have only been included on a selection of datasets (TUD#3 microcline, LD1
microcline, BCS 375 albite and 67796b plagioclase). The error bars shown are typical.

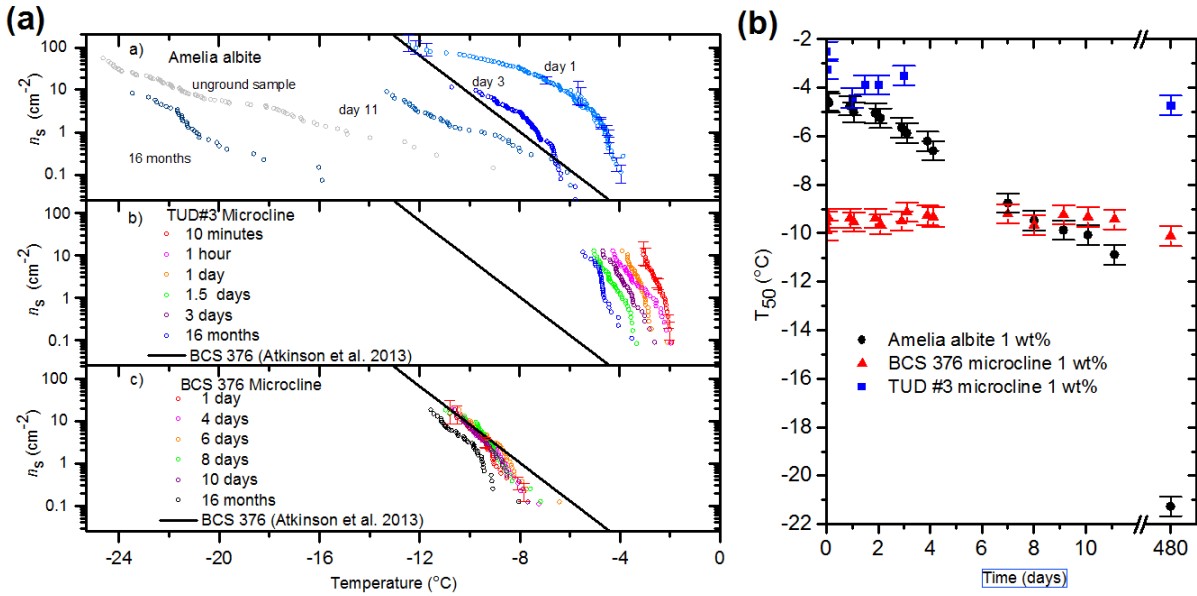

**Figure 4:** (a) The dependence of $n_s$ on time spent in water for three feldspar samples. The
time periods indicate how long samples were left in contact with water. Fresh samples were
tested minutes after preparation of suspensions. Note that ice nucleation temperatures of BCS
376 are almost the same after 16 months in water while those of Amelia albite decreases by
around 16°C. TUD #3 microcline loses activity quickly in the first couple of days of exposure
to water but total decrease in nucleation temperatures after 16 months is only around 2°C. (b)
Median freezing temperature against time left in suspension for BCS 376 microcline, TUD#3
microcline and Amelia albite.

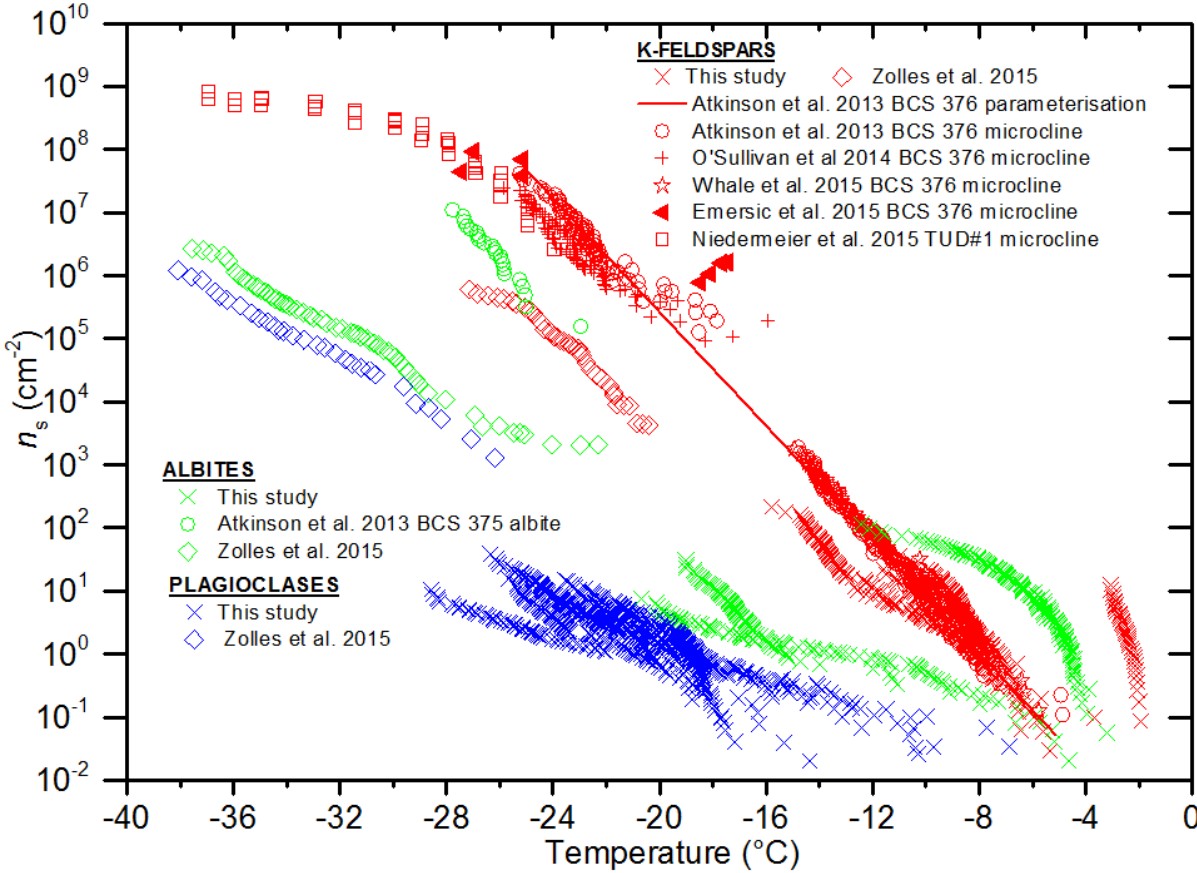

**Figure 5**: Comparison of literature data from Atkinson et al. (2013), Emersic et al. (2015), Niedermeier et al. (2015) and Zolles et al. (2015) with data from this study. Feldspars are coloured according to their composition, as in Figure 3. 0.1 wt% data for Amelia albite and LD1 microcline, which is not shown in figure 3, has been included. Where samples are known to lose activity with time the most active runs have been shown. Note that data from Niedermeier et al. (2015) includes some data from Augustin-Bauditz et al. (2014).

Table 1. Plagioclase feldspars used in this study.

| Sample | Composition* | Source location | Source of composition/phase data | Space group | Point group | Crystal system |
|---|---|---|---|---|---|---|
| Anorthite glass | $An_{100}$ | Synthetic sample | (Carpenter, 1991) | - | - | - |
| ANC 68 | $An_{100}$ | Synthetic sample | (Carpenter, 1991) describes similar feldspars | $P\bar{1}$ | $\bar{1}$ | triclinic |
| 148559 | $An_{99.5}Ab_{0.5}$ | University of Cambridge mineral collection | ----- | $P\bar{1}$ | $\bar{1}$ | triclinic |
| 21704a | $An_{86}Ab_{14}$ | Viakfontein, Bushveld complex, Transvaal (Harker collection no. 21704) | (Carpenter et al., 1985) | $P\bar{1}$-$I\bar{1}$ | $\bar{1}$ | triclinic |
| Surt M | $An_{64}Ab_{36}$ | Surtsey (no. 7517, Iceland Natural History Museum) Phenocrysts from volcanic ejecta | (Carpenter, 1986) | $C\bar{1}$ | $\bar{1}$ | triclinic |
| 67796b | $An_{60}Or_1Ab_{39}$ | Gulela Hills, Tanzania (Harker collection no. 67796) | (Carpenter et al., 1985) | Incommen-surate order | - | triclinic |
| 97490 | $An_{27}Or_1Ab_{71}$ | Head of Little Rock Creek, Mitchell co., N. Carolina (P. Gay, U.S.N.M. no. 97490) | (Carpenter et al., 1985) | Incommen-surate order | - | triclinic |

*This refers to the chemical makeup of the feldspars. An stands for anorthite, the calcium end-
member, Ab stands for albite, the sodium end-member and Or stands for orthoclase, the potassium
end-member.
Table 2. Alkali feldspars used in this study.

| Sample | Dominant feldspar phase | Source location | Source of composition/phase data | Space group | Point group | Crystal system |
|---|---|---|---|---|---|---|
| LD1 microcline | microcline | University of Leeds rock collection | XRD | C$\bar{1}$ | $\bar{1}$ | triclinic |
| LD2 sanidine | sanidine | University of Leeds rock collection | XRD | C2/$m$ | 2/$m$ | monoclinic |
| LD3 microcline | microcline | University of Leeds rock collection | XRD | C$\bar{1}$ | $\bar{1}$ | triclinic |
| BCS 376 microcline | microcline | Bureau of Analysed Samples Ltd | Reference sample/XRD (Atkinson et al., 2013) | C$\bar{1}$ | $\bar{1}$ | triclinic |
| Amelia Albite (un-ground) | albite | Amelia Courthouse, Amelia Co., Virginia (Harker mineral collection) | (Carpenter et al., 1985) | C$\bar{1}$ | $\bar{1}$ | triclinic |
| Amelia Albite ground | albite | Amelia Courthouse, Amelia Co., Virginia (Harker mineral collection) | (Carpenter et al., 1985) | C$\bar{1}$ | $\bar{1}$ | triclinic |
| TUD#1 microcline | microcline | Minas Gerais, Brazil | XRD | C$\bar{1}$ | $\bar{1}$ | triclinic |
| TUD#2 albite | albite* | Norway | XRD | C$\bar{1}$ | $\bar{1}$ | triclinic |
| TUD#3 microcline | microcline | Mt. Maloso, Malawi | XRD | C$\bar{1}$ | $\bar{1}$ | triclinic |
| BCS 375 albite | albite | Bureau of Analysed Samples Ltd | Reference sample/XRD (Atkinson et al., 2013) | C$\bar{1}$ | $\bar{1}$ | triclinic |

* We note that the XRD pattern was also consistent with oligoclase, which is close to albite in
composition. The identification of albite is consistent with that of Alexei Kiselev (Personal
communication).