# Peer review of "Not all feldspars are equal: a survey of ice nucleating"

_Atmospheric Chemistry and Physics, 2016_

## Referee Comment (RC1) · Anonymous Referee #1 · 22 Mar 2016

The present study provides an overview of the ice nucleation activity of different feldspars varying in their chemical composition and crystal structure. In total 15 different feldspars including plagioclase and alkali feldspars were analyzed for one weight percentage using a special freezing array method called $\mu$l-NIPI (microliter Nucleation by Immersed Particle Instrument). Additionally, the samples were characterized with respect to their BET surface area and mineralogical structure at least for the dominating feldspar phase. It was found that K-feldspars generally nucleate ice more efficiently than other types of feldspar except for the two most efficient so-called "hyper-active" alkali feldspars (one of microcline and albite). One of the main statements of the study is that the observed ice nucleation efficiency of different feldspars varies. This effect was less pronounced in the case of K-feldspars. I consider this as the key information of this study. The influence of particle aging in water on the ice nucleation ability of

selected feldspars was investigated, too. Intensive change was found for albite one of the hyper-active feldspars over 16 months while other feldspars show no significant change. Ice nucleation surface site densities were derived from the data and compared to data available in literature. The paper is comprehensible and nicely written. It addresses scientific questions in the scope of ACP, but I have some major concerns about scientific quality. I recommend publication in ACP after discussing and incorporating the general and specific comments thoroughly.

General comments

(1) To analyze and characterize the chemical composition and crystallographic structure and its features more effort has to be done as realized in the present study. This fact is also explicitly stated by the authors. However, this study only present a starting point as the feldspar characterization method used in the present study is imprecise. Hence the ice nucleation ability of the different feldspars could not be related to e.g. special chemical or crystallographic features such as intracrystalline defect or any other intrinsic property. This would be of great interest. The mineralogical composition is given in Table 1 and for the dominant feldspar phase in Table 2. I am wondering if any information of the general composition (e.g. any component but feldspar) or crystal purity is available or measureable. One main result from the present study is that the ice nucleation ability varies for feldspar except for K-feldspar. What is the reason for that? Does it depend also on the source regions for the different feldspars? Would it be necessary for modellers to account for such an effect? This should be first discussed and second stated more clearly in the manuscript.

(2) The authors try to explain the observed freezing behaviour of pure feldspars and feldspars aged in water using the concept of “active sites”. It is unclear how exactly an active site is defined in the context it is used in the present study. Rather it seems that an active site is used as a construct with which almost anything can be explained if it is not related to a property of an ice nucleating particle, which should be determined from an independent measurement, see general comment (1). The concept of active sites

has to be introduced and motivated earlier in the paper and caution is required when conclusions are drawn. On page 5 line14-17, the authors distinguish between 3 different kinds of active site. Is it known for example whether the active site type (i) is similar for all feldspars? Or do they differ? This is not clear to me. The study includes many interesting indications what these special properties are or at least how they change for different types of feldspars or due to aging in water, but a clear comprehensive explanation is missing.

(3) At the moment without further tests, I am not convinced that the "hyper-activity" of one of the microcline and albite samples is real. I think the contribution from to biological ice nucleator contaminants can not be ruled out completely. The feldspar suspensions were left at room temperature (p.6 l.22). Under such conditions biological activity is not suppressed. The relatively gentle treatment in 100°C water for 15 min might destroy proteins (p.9 l.24-26) but probably not ice active polysaccharides or other organic/biological substances ice active at quite high temperature (Pummer et al., 2012; Tobo et al., 2014; O'Sullivan et al., 2014). Additional treatments with H2O2, H2SO4, etc. could be helpful as only heating is not always enough.

(4) Figure 3 and 4 has very poor quality and presents similar statements. The symbols are partly not visible. One can not distinguish between the different times of aging. The measurement uncertainties are missing, which must be known to judge if it is a real trend or just uncertainty of the experiment. This is most obvious in Fig.3 panel b) and c). I suggest to combine these figures.

Specific comments

p.1 l.16 & p.2 l.3 Specify what is meant by "soil dust". In the context of this study it seems that you mean mineral dust. Otherwise you need to clarify this statement.

p.1 l.22 Considering point 1 and 2 of my major concerns I don't agree that the feldspar samples are "well-characterized". They are only characterized with respect to their "macroscopic crystal structure" and BET surface area, but other probably decisive

properties such as intra-crystalline defects are not considered at all.

p.5 l.1-3 It might be true that the knowledge about the chemical composition of airborne mineral dust is limited. However, to be fair some studies investigating the composition of atmospheric mineral dust can be mentioned for example Glaccum and Prospero (1980), Kandler et al. (2007, 2009) to name a few.

p.5 l.24-27 and in all figures The labelling of the different feldspar samples might be precise and traceably, but totally confusing for the reader. I suggest to simplify the nomenclature in a way that it will be transparent for the reader to whom the paper is addressed. Probably a table in the Appendix could be useful for the precise description.

p.5 l.28-30, p.6 l.1-2 Please give a motivation for this treatment of the sample.

p. 6 l.10-14 In that context, Zolles et al, (2015) claimed that grinding could lead to a disclosing of active sites and even an enhancement of available active sites. This should be addressed in the manuscript as well.

p.7 l.3 Specify what is meant by “quantity”. Is this quantity related to mass or mass concentration?

p.7 l.5-7 I do not understand why using a small dry nitrogen flow should prevent frozen droplets from affecting their neighbour liquid droplets?

p.7 l.7 How large are the droplets and do they vary in size? How many feldspar particles are immersed in the droplets and how does this number vary between different droplets. Statements to clarify these quantities are required. These are necessary information the reader needs to assess the reasonability of the approach of determining the uncertainty in ns.

p.8 l.2-16 As mentioned in the last section further information of the distribution of particles (hence potential active sites) over the droplet population are needed. I do not understand the procedure to derive the uncertainty in ns. Especially I could not follow how the two distributions are combined. Further, is the assumption ” that each droplet

contains a representative surface area distribution" justified? Maybe this is only true for rather high particle mass concentrations.

p.10 l. 5-9 I do not agree completely. I agree that flat increase in ns indicates a diversity of ice nucleation properties. However, the steeper slopes of ns or analogous quantities at higher temperature are also predicted by Classical Nucleation Theory assuming only one contact angle. In other words, even when similar ice nucleation properties (one contact angle) are assumed, the slopes become steeper at higher temperature. As a conclusion, the same effect can be explained also by a different hypothesis.

p.11 l.19-20 It seems that in Fig.3 the variation might also be explained by the measurement uncertainty? There is no trend in one direction with increasing time, or is the legend incorrectly labelled? This must be clarified.

p.11 l. 25-28 Additionally, the study of Marcolli et al. (2007) and Hartmann et al. (2016) can be mentioned.

p.13 l.26-32 Larger ns values always imply lower available particle surface area relevant for heterogeneous ice nucleation or lower number of active sites when similar ice nucleating materials are analyzed. The mass concentration and size of a droplet containing particles of different sizes (particle distribution) is not the essential quantity, but the total surface area of the particles. I feel that in both experiments (Atkinson et al., 2013 and Zolles et al., 2015) this quantity is not determined with sufficient reliability. Consequently, this is not a conclusive argument. If ns is carefully derived from the experimental data, this effect should be already considered. Error bars could clarify the uncertainty in ns derived from different experiment.

p.23 Fig.2 For the observed freezing at low temperature, it is necessary to show the limit of the measurement due to pure water freezing.

p.23 l.12 I do not agree that ns values can be simply subtracted without introducing further unnecessary uncertainty. An internal mixture of different ice nucleating particles

has to be accounted for.

Technical corrections

p.3 l.16-18 The citation list in brackets has to be sorted in the order beginning from the oldest to the recent publications. This needs to be changed in the whole manuscript.

p.16 l.2 . . . higher temperatures . . .

Bibliography

Atkinson, J. D., B. J. Murray, M. T. Woodhouse, T. F. Whale, K. J. Baustian, K. S. Carslaw, S. Dobbie, D. O'Sullivan, and T. L. Malkin (2013), The importance of feldspar for ice nucleation by mineral dust in mixed-phase clouds, Nature, 498(7454), 355-358, doi:10.1038/nature12278.

Glaccum, R. A., and J. M. Prospero (1980), Saharan aerosols over the tropical North-Atlantic - mineralogy, Mar. Geol., 37(3-4), 295-321, doi:10.1016/0025-3227(80)90107-3.

Hartmann, S., H. Wex, T. Clauss, S. Augustin-Bauditz, D. Niedermeier, M. Rösch, and F. Stratmann (2016), Immersion Freezing of Kaolinite: Scaling with Particle Surface Area, J. Atmos. Sci., 73, 263-278, doi:10.1175/JAS-D-15-0057.1.

Hoose, C., and O. Möhler (2012), Heterogeneous ice nucleation on atmospheric aerosols: a review of results from laboratory experiments, Atmos. Chem. Phys., 12(20), 9817-9854, doi:10.5194/acp-12-9817-2012.

Kandler, K., N. Benker, U. Bundke, E. Cuevas, M. Ebert, P. Knippertz, S. Rodriguez, L. Schuetz, and S. Weinbruch (2007), Chemical composition and complex refractive index of Saharan Mineral Dust at Izana, Tenerife (Spain) derived by electron microscopy, Atmos. Environ., 41(37), 8058-8074, doi:10.1016/j.atmosenv.2007.06.047.

Kandler, K., et al. (2009), Size distribution, mass concentration, chemical and mineralogical composition and derived optical parameters of the boundary layer

aerosol at Tinfou, Morocco, during SAMUM 2006, Tellus, Ser. B, 61(1), 32-50, doi:10.1111/j.1600-0889.2008.00385.x.

O'Sullivan, D., B. J. Murray, T. L. Malkin, T. F. Whale, N. S. Umo, J. D. Atkinson, H. C. Price, K. J. Baustian, J. Browse, and M. E. Webb (2014), Ice nucleation by fertile soil dusts: relative importance of mineral and biogenic components, Atmos. Chem. Phys.„ 14(4), 1853-1867, doi:10.5194/acp-14-1853-2014.

Marcolli, C., S. Gedamke, T. Peter, and B. Zobrist (2007), Efficiency of immersion mode ice nucleation on surrogates of mineral dust, Atmos. Chem. Phys., 7(19), 5081-5091, doi:10.5194/acp-7-5081-2007.

Pummer, B. G., H. Bauer, J. Bernardi, S. Bleicher, and H. Grothe (2012), Suspendable macromolecules are responsible for ice nucleation activity of birch and conifer pollen, Atmos. Chem. Phys., 12(5), 2541-2550, doi:10.5194/acp-12-2541-2012.

Tobo, Y., P. J. DeMott, T. C. J. Hill, A. J. Prenni, N. G. Swoboda-Colberg, G. D. Franc, and S. M. Kreidenweis (2014), Organic matter matters for ice nuclei of agricultural soil origin, Atmos. Chem. Phys., 14(16), 8521-8531, doi:10.5194/acp-14-8521-2014.

Zolles, T., J. Burkart, T. Haeusler, B. Pummer, R. Hitzenberger, and H. Grothe (2015), Identification of Ice Nucleation Active Sites on Feldspar Dust Particles, J. Phys. Chem. A, 119(11), 2692-2700, doi:10.1021/jp509839x.

---

## Referee Comment (RC2) · Anonymous Referee #2 · 22 Mar 2016

Harrison et al. present immersion freezing measurements of 15 different feldspar mineral samples, using a droplet freezing technique. Their intent is to explore the variability of ice nucleation properties of the feldspar mineral group, and attempt to relate this to the mineralogical properties. They also report that one of three feldspar samples tested exhibited significant and rapid loss of ice activity as the mineral sample sat in water for hours or months. As the feldspar minerals have recently been identified as the most efficient mineral ice nucleants, this topic will be interesting to this journal's readership. The paper would be a lot stronger if more details regarding the precise mineralogical properties and crystal/lattice structure of the different minerals was provided. As written there is little deeper or fundamental insight provided regarding the quite different freezing properties of the various minerals and how this relates to the actual mineral properties. The large change, or lack of change, in freezing ability with time spent in

water is a valuable finding, though it was not clear why those 3 mineral samples were the only ones tested for the effects of time spent in water. While not a great deal of new scientific information or insights are presented here, this paper still provides some valuable new data regarding an important atmospheric ice nucleant. This manuscript should be suitable for publication in ACP, but the following issues need to be addressed first.

The information provided on the different mineralogical compositions of the feldspar minerals is good, and Figure 1 summarizes this well. I was surprised to not also see more details provided regarding the crystal lattice structure, symmetry, and space group of the different mineral phases. These properties are often referred to in the text to try to understand the observed ice nucleation properties, but without a table or figure summarizing this information it hard to understand this important aspect. Please add as much detail regarding the other known properties of these feldspar minerals in a well organized table or similar.

Introduction: It seems that the recent effort by Perlwitz et al. to incorporate better representations of the variable mineralogical composition of dust into global models should be referred to here. A major challenge regarding understanding and predicting the ice nucleation properties of atmospheric mineral particles is that we do not have a good understanding of the distribution, abundance, and transport of the different mineral types in the atmosphere.

Perlwitz, J. P.; Pérez García-Pando, C.; Miller, R. L. Predicting the mineral composition of dust aerosols – Part 1: Representing key processes. Atmos. Chem. Phys. 2015, 15, 11593–11627, doi:10.5194/acp-15-11593-2015.

On a related note, it is important to also have some discussion of the size of atmospheric feldspar mineral particles. What aerosol size mode are these typically found in? This is crucial to predict their transport, lifetime, and deposition. One of the reasons that the clay minerals have been focused on for so long is that they tend to be present

in the smaller atmospheric mineral particle sizes, and thus have longer lifetimes.

Although you focus on immersion freezing here – without actually stating the heterogeneous ice nucleation mode you measure here (please clarify this so it is clear to the reader), it was odd that this paper on the depositional ice nucleation properties of Feldspar was not cited:

Yakobi-Hancock, J. D.; Ladino, L. A.; Abbatt, J. P. D. Feldspar minerals as efficient deposition ice nuclei. Atmos. Chem. Phys. 2013, 13, 11175–11185, doi:10.5194/acp-13-11175-2013.

Some discussion of the vulnerability of feldspar minerals to chemical attack by e.g. sulfuric acid should be included. This is quite important to understand the actual contribution of feldspar minerals to atmospheric ice nucleation, and also provides some insights into the nature of the ice active sites. Wex et al. (2014), already cited here, discuss some of these aspects. I believe it is well known in the mineralogy community that these feldspar minerals can be readily converted to amorphous clay structure through reaction with sulfuric acid.

Page 6: Why were those 3 mineral samples selected out of the 15 to perform the extended time in water experiments on? It would be valuable to conduct these tests on a larger number of the minerals, since the behavior seems quite variable between minerals. At the least some justification for the 3 sample chosen could be given.

Page 6, line 31: Citing 8 of the authors own publications that uses the same (and rather simple) experimental method seems like excessive and unnecessary self-citation, especially when Whale et al. (2015a) already provides a detailed discussion of the method. Please restrict these to the most necessary and relevant citations.

Page 7, line 19: More of the recently published experimental work that has explored the role of time-dependent freezing should be cited, such as:

Wright, T. P.; Petters, M. D. The role of time in heterogeneous freezing nucleation. J.

Geophys. Res. Atmos. 2013, 118, 3731–3743, doi:10.1002/jgrd.50365.

Some discussion of the similarity of the Monte Carlo approach to estimate the uncertainty of the n_s values to other work should be presented. Is this the first time these authors have used this approach, or that anyone has used a similar approach? Wright & Petters (JGR, 2013; cited above) also used a Monte Carlo approach to analyze and interpret their droplet freezing data. Please discuss this. As it is presented it reads as if this is a completely new approach.

Page 8, line 14: "We assume that each droplet contains a representative surface area distribution." Please clarify what "representative" means. Is the goal to account for the non-uniform distribution of particle number and surface area in each droplet?

Figure 3 is hard to read at the presented size. The symbols are too small and faint.

Figure 4 is begging for some error bars or other measurement of the uncertainties, so it can be determined what degree of the observed changes in median freezing temperature are significant and above the experimental uncertainties. It seems that only the Amelia albite sample displayed any significant change.

Figure 5: Some annotations/captions added directly to the figure pointing out what data is plotted where would improve the clarity of this paper. It is difficult to have to keep going back to the figure legend to decode what each dataset is from.

Technical corrections:

Page 3, line 19: "experiments"

Page 7, line 5: Missing a space, should be "C min-1"

Page 9, line 30: "(nucleation rate) vs." Versus what?

A space in-between the number and "degreeC" is often missing, such as throughout pages 12 & 13.

Page 14, line 18: "regimen this study". Word is missing?

Page 15, line 13: "sites"

Page 15, line 15: "that are stable"

---

## Author Comment (AC1) · 20 Jun 2016

Response to referee 1

We would like to thank the reviewer for their valuable comments. We have reworked the paper to address the relevant issues where necessary. The reviewer comments are written in italics, our response in normal type and changes to the manuscript in bold.

General comments

(1) *To analyze and characterize the chemical composition and crystallographic structure and its features more effort has to be done as realized in the present study. This fact is also explicitly stated by the authors. However, this study only present a starting point as the feldspar characterization method used in the present study is imprecise. Hence the ice nucleation ability of the different feldspars could not be related to e.g. special chemical or crystallographic features such as intracrystalline defect or any other intrinsic property. This would be of great interest. The mineralogical composition is given in Table 1 and for the dominant feldspar phase in Table 2. I am wondering if any information of the general composition (e.g. any component but feldspar) or crystal purity is available or measureable. One main result from the present study is that the ice nucleation ability varies for feldspar except for K-feldspar. What is the reason for that? Does it depend also on the source regions for the different feldspars? Would it be necessary for modellers to account for such an effect? This should be first discussed and second stated more clearly in the manuscript.*

We think the paper is very important since it is the first survey across the feldspar group and it indicates that specific feldspars nucleate ice more efficiently than others. This means we are now in a much better position to do much more focused and detail orientated future studies where we look at specific properties of the select feldspars which nucleate ice effectively. Much of what the referee suggests in terms of trying to find what intrinsic property controls nucleation is sensible, but we view a detailed study as a next step. Nevertheless, the present results do indicate that the strongest ice nucleation is limited to the alkali feldspars and we hypothesise in the paper that the nucleating ability is related to microtexture. We are currently working on this hypothesis, but characterising microtexture and doing controlled experiments is a major study and is very much the topic of a future paper.

As the referee suggests, there are other properties of the feldspars which could be of interest. We have incorporated the space and point groups as well as the source locations of the minerals into tables 1 and 2 to try better display the known information for the studied minerals.

*(2) The authors try to explain the observed freezing behaviour of pure feldspars and feldspars aged in water using the concept of "active sites". It is unclear how exactly an active site is defined in the context it is used in the present study. Rather it seems that an active site is used as a construct with which almost anything can be explained if it is not related to a property of an ice nucleating particle, which should be determined from an independent measurement, see general comment (1). The concept of active sites has to be introduced and motivated earlier in the paper and caution is required when conclusions are drawn. On page 5 line14-17, the authors distinguish between 3 different kinds of active site. Is it known for example whether the active site type (i) is similar for all feldspars? Or do they differ? This is not clear to me. The study includes many interesting indications what these special properties are or at least how they change for different types of feldspars or due to aging in water, but a clear comprehensive explanation is missing.*

We have expanded our discussion of active sites in the experimental section:

**To allow comparison of the ability of different materials to nucleate ice, the number of active sites is normalised to the surface area available for nucleation. This yields the ice nucleation active site density, $n_s(T)$. $n_s(T)$ is the number of ice nucleating sites that become active per surface area on cooling from 0°C to temperature $T$ and can be calculated using (Connolly et al., 2009):**

$$\frac{n(T)}{N} = 1 - \exp(-n_s(T)A) \qquad\qquad (1)$$

**Where $n(T)$ is the number of droplets frozen at temperature $T$, $N$ is the total number of droplets in the experiment and $A$ is the surface area of nucleator per droplet.**

**Active sites may be related to imperfections in a crystal structure, such as cracks or defects, or may be related to the presence of quantities of other more active materials located in specific locations at a surface. While the fundamental nature of sites is not clear, and may be different for different materials, $n_s$ is a pragmatic parameter which allows us to empirically define the ice nucleating efficiency of a range of materials (Vali, 2014).**

**This description is site specific and does not include time dependence. The role of time dependence in ice nucleation has recently been extensively discussed (Vali, 2014;Vali et al., 2014;Vali, 2008;Herbert et al., 2014;Wright et al., 2013). For feldspar (at least for BCS 376 microcline) it is thought that the time dependence of nucleation is relatively weak and that the particle to particle, or active site to active site, variability is much**

**more important (Herbert et al., 2014). The implication of this is that specific sites on the surface of most nucleators, including feldspars, nucleate ice more efficiently than the majority of the surface. As this study is aimed at comparing and assessing the relative ice nucleating abilities of different feldspars we have not determined the time dependence of observed ice nucleation in this work, although this would be an interesting topic for future study.**

We acknowledge the referee's comments on whether the active site type (i) is similar for all feldspars and on the absence of 'a clear comprehensive explanation' with reference to the types of site. This is a problem for the entire field of ice nucleation, there is no clear, comprehensive and generally agreed upon explanation for why any substance should nucleate ice. We have made some intriguing observations which move us in the direction of a more comprehensive understanding. In the text we already discuss the different characteristics of sites on different feldspars samples. In addition we have expanded on the discussion in the conclusions section with the following lines:

**'It is possible that the sites of type $i$ are present on the typical K-feldspars, but we do not observe them because ice nucleates on more active sites. Whether these different sites are all related to similar features on the surfaces or if they are each related to different types of features is not known. Nevertheless, it appears that feldspars are characterised by a range of site types with varying stability and activity.'**

*(3) At the moment without further tests, I am not convinced that the "hyper-activity" of one of the microcline and albite samples is real. I think the contribution from to biological ice nucleator contaminants can not be ruled out completely. The feldspar suspensions were left at room temperature (p.6 l.22). Under such conditions biological activity is not suppressed. The relatively gentle treatment in 100◦C water for 15 min might destroy proteins (p.9 l.24-26) but not ice active polysaccharides or other organic/biological substances ice active at quite high temperature (Pummer et al., 2012; Tobo et al., 2014; O'Sullivan et al., 2014). Additional treatments with H2O2, H2SO4, etc. could be helpful as only heating is not always enough.*

As the referee states the treatment of the sample at 100ºC would destroy proteins but potentially not other organic substances. However the only biological substances we know to be active at the high temperatures displayed for the TUD#3 microcline and Amelia albite are proteins. This leads us to think that the activity is inherent to the feldspars. Additional treatments with $H_2O_2$ or $H_2SO_4$ might be helpful but may also influence ice nucleation by feldspars which would make any results difficult to interpret. We agree that we should be more cautious in our statements and have modified the text accordingly:

**'Certain biological nucleators have been observed to retain their ice nucleating activity despite heat treatment of this type (Pummer et al., 2012;O'Sullivan et al., 2014;Tobo et al., 2014) however, to the best of our knowledge, no biological species has been observed to nucleate ice at such warm temperatures after heat treatment. Additionally, grinding of Amelia albite which had been stored as a powder for many years increases its ice nucleating potential, which is consistent with exposing fresh surfaces with features which decay away on contact with water. This behaviour is not consistent with biological nucleators, unless the biological entity is within the Amelia albite particles and is somehow dispersed through the particle population during grinding. While we cannot exclude the possibility that some unknown biological species is present on microcline TUD#3 and Amelia albite it seems more likely that the minerals themselves are responsible for the observed ice nucleation activity.'**

*(4) Figure 3 and 4 has very poor quality and presents similar statements. The symbols are partly not visible. One can not distinguish between the different times of aging. The measurement uncertainties are missing, which must be known to judge if it is a real trend or just uncertainty of the experiment. This is most obvious in Fig.3 panel b) and c). I suggest to combine these figures.*

We have combined both plots into a single figure with multiple panels. We think both panels are needed to emphasise the nature of the decay in activity over time. It is noted that the figures are difficult to see due to resizing for the discussions paper. This should improve in the final copy. In addition, we have used stronger colours for the points and error bars as well as expanding the x-axis.

Specific comments

*p.1 l.16 & p.2 l.3 Specify what is meant by "soil dust". In the context of this study it seems that you mean mineral dust. Otherwise you need to clarify this statement.*

The term soil dusts has been removed from the text to prevent any confusion.

*p.1 l.22 Considering point 1 and 2 of my major concerns I don't agree that the feldspar samples are "well-characterized". They are only characterized with respect to their "macroscopic crystal structure" and BET surface area, but other probably decisive properties such as intra-crystalline defects are not considered at all.*

We have replaced the phrase "well characterised" with simply "characterised".

*p.5 l.1-3 It might be true that the knowledge about the chemical composition of airborne mineral dust is limited. However, to be fair some studies investigating the composition of atmospheric mineral dust can be mentioned for example Glaccum and Prospero (1980), Kandler et al. (2007, 2009) to name a few.*

We have now cited these papers in the relevant section and the text now reads:

**'There is limited information about the composition of airborne atmospheric mineral dusts (Glaccum and Prospero, 1980;Kandler et al., 2007;Kandler et al., 2009); where mineralogy is reported the breakdown of the feldspar family has only been done in a limited way.'**

*p.5 l.24-27 and in all figures The labelling of the different feldspar samples might be precise and traceably, but totally confusing for the reader. I suggest to simplify the nomenclature in a way that it will be transparent for the reader to whom the paper is addressed. Probably a table in the Appendix could be useful for the precise description.*

We have attempted to use naming conventions compatible with the geology literature. Our samples are named in the same fashion as they were in the studies which initially characterised them, where possible. Where we have introduced new samples we have followed the same naming conventions. We understand that the

number of samples and their naming can be cumbersome but we feel that it is important to retain consistency with previous work. We would therefore prefer to use the current nomenclature.

*p.5 l.28-30, p.6 l.1-2 Please give a motivation for this treatment of the sample*

We have inserted:

**'As these two samples are chemically identical, differing only in that one is amorphous and the other crystalline, comparison of the ice nucleating efficiency of the two samples has the potential to reveal information about the impact of feldspar crystal structure on ice nucleating efficiency.'**

*p. 6 l.10-14 In that context, Zolles et al, (2015) claimed that grinding could lead to a disclosing of active sites and even an enhancement of available active sites. This should be addressed in the manuscript as well.*

The activity increase seen by Zolles *et al,* (2015) has now been mentioned:

**Zolles at al. (2015) have suggested that grinding can lead to active sites being revealed, or the enhancement of existing active sites. It was shown in Whale et al. (2015) that differently ground samples of BCS 376 microcline nucleate ice similarly. In contrast Hiranuma et al. (2014) show that ground hematite nucleates ice more efficiently (normalised to surface area) than cubic hematite. The evidence suggests that the ice nucleating efficiencies of different materials respond differently to grinding processes.**

*p.7 l.3 Specify what is meant by "quantity". Is this quantity related to mass or mass concentration?*

This has now been adjusted to:

**"Briefly, droplets of an aqueous suspension, containing a known mass concentration of feldspar particles are pipetted onto a hydrophobic coated glass slide."**

*p.7 l.5-7 I do not understand why using a small dry nitrogen flow should prevent frozen droplets from affecting their neighbour liquid droplets?*

This has been further clarified in the text as follows:

**"This slide is placed on a temperature controlled stage and cooled from room temperature at a rate of 5 °C min⁻¹ to 0 °C and then at 1 °C min⁻¹ until all droplets are frozen. Dry nitrogen is flowed over the droplets at 0.2 l min⁻¹ to prevent frozen droplets from affecting neighbouring liquid droplets. Whale et al. (2015) demonstrated that a dry nitrogen flow prevents condensation and frost accumulating on the glass slide so ice from a frozen droplet cannot trigger freezing in neighbouring droplets."**

*p.7 l.7 How large are the droplets and do they vary in size? How many feldspar particles are immersed in the droplets and how does this number vary between different droplets. Statements to clarify these quantities are required. These are necessary information the reader needs to assess the reasonability of the approach of determining the uncertainty in ns.*

The droplets are of 1 ± 0.025 μl volume, we have inserted the following (the uncertainty is small because we use an electronic pipette with low uncertainty):

**'Briefly, 1 ± 0.025 μl droplets of an aqueous suspension, containing a known mass concentration of feldspar particles are pipetted onto a hydrophobic coated glass slide.'**

We comment on the number of particles per droplet in the next comment

*p.8 l.2-16 As mentioned in the last section further information of the distribution of particles (hence potential active sites) over the droplet population are needed. I do not understand the procedure to derive the uncertainty in ns. Especially I could not follow how the two distributions are combined. Further, is the assumption " that each droplet contains a*

*p.7 l.5-7 I do not understand why using a small dry nitrogen flow should prevent frozen droplets from affecting their neighbour liquid droplets?*

This has been further clarified in the text as follows:

**"This slide is placed on a temperature controlled stage and cooled from room temperature at a rate of 5 °C min$^{-1}$ to 0 °C and then at 1 °C min$^{-1}$ until all droplets are frozen. Dry nitrogen is flowed over the droplets at 0.2 l min$^{-1}$ to prevent frozen droplets from affecting neighbouring liquid droplets. Whale et al. (2015) demonstrated that a dry nitrogen flow prevents condensation and frost accumulating on the glass slide so ice from a frozen droplet cannot trigger freezing in neighbouring droplets."**

*p.7 l.7 How large are the droplets and do they vary in size? How many feldspar particles are immersed in the droplets and how does this number vary between different droplets. Statements to clarify these quantities are required. These are necessary information the reader needs to assess the reasonability of the approach of determining the uncertainty in ns.*

The droplets are of 1 ± 0.025 μl volume, we have inserted the following (the uncertainty is small because we use an electronic pipette with low uncertainty):

**'Briefly, 1 ± 0.025 μl droplets of an aqueous suspension, containing a known mass concentration of feldspar particles are pipetted onto a hydrophobic coated glass slide.'**

We comment on the number of particles per droplet in the next comment

*p.8 l.2-16 As mentioned in the last section further information of the distribution of particles (hence potential active sites) over the droplet population are needed. I do not understand the procedure to derive the uncertainty in ns. Especially I could not follow how the two distributions are combined. Further, is the assumption " that each droplet contains a*

*representative surface area distribution" justified? Maybe this is only true for rather high particle mass concentrations.*

We have inserted the following to address the question of the number of particles per droplet:

**'By assuming that the BET surface area of the feldspar powders is made up of monodisperse particles it can be estimated that droplets containing 1 wt% of feldspar will each contain around $10^6$ particles. While there will be a distribution of particle sizes we assume that there are enough particles per droplet that the uncertainty in surface area per droplet due to the distribution of particles through the droplets is negligible. In contrast, it has been suggested that ice nucleation data could be explained by variability of nucleator surface area through the droplet population (Alpert and Knopf, 2016). Our assumption that each droplet contains a representative surface area is supported by our previous work where we show that $n_s$ derived from experiments with a range of feldspar concentrations are consistent with one another(Atkinson et al., 2013) (Atkinson et al., 2013; Whale et al., 2015). If the particles were distributed through the droplets in such a way that some droplet contained a much larger surface area of feldspar than others we would expect the slope of $n_s$ with temperature to be artificially shallow. The slope would be artificially shallow because droplets containing more than the average feldspar surface area would tend to freeze at higher temperatures and vice versa. However, the fact that $n_s$ data for droplets made from suspensions made up with a wide range of different feldspar concentrations all line up shows that the droplet to droplet variability in feldspar surface area is minor (Atkinson et al., 2013;Whale et al., 2015). Hence, the droplet to droplet variability in feldspar surface area is neglected and the uncertainty in surface area per droplet in these experiments is estimated from the uncertainties in weighing, pipetting and specific surface area of the feldspars.**

With regard to the description of the way the uncertainty in $n_s$ is calculated we have changed the description of the method in an effort to make it clearer. It now reads:

**'In order to estimate the uncertainty in $n_s(T)$ due to the randomness of the distribution of the active sites in droplet freezing experiments, we conducted Monte Carlo simulations. Wright and Petters (2013) previously adopted a similar approach to simulate the distribution of active sites in droplet freezing experiments.  In these**

simulations, we generate a list of possible values for the number of active sites per droplet ($\mu$). The theoretical relationship between the fraction of droplets frozen and $\lambda$ can be derived from the Poisson distribution:

$$\frac{n(T)}{N} = 1 - \exp(-\mu) \tag{2}$$

The simulation works in the following manner. First, we take a value of $\mu$ and we simulate a corresponding random distribution of active sites through the droplet population for an experiment. Every droplet containing one or more active sites is then considered to be frozen. In this way, we can obtain a simulated value of the fraction frozen for a certain value of $\mu$. Repeating this process many times and for all the possible values of $\mu$, we obtain a distribution of possible values of $\mu$ that can explain each value of the observed fraction frozen. This resulting distribution is neither Gaussian nor symmetric, so in order to propagate the uncertainty to $n_s(T)$ values, we take the following steps. First, we generate random values of $\mu$ following the corresponding previously simulated distribution for each value of the fraction frozen. Then, we simulate random values of $A$ following a Gaussian distribution centred on the value derived from the specific surface area per droplet with the standard deviation derived from the uncertainty in droplet volume and specific surface area. We assume that each droplet contains a representative surface area distribution as discussed above. This process results in two distributions, one for $A$ and one for $\mu$, with these distributions we can calculate the resultant distribution of $n_s(T)$ values, and from that distribution, we obtain the 95% confidence interval.'

*p.10 l. 5-9 I do not agree completely. I agree that flat increase in ns indicates a diversity of ice nucleation properties. However, the steeper slopes of ns or analogous quantities at higher temperature are also predicted by Classical Nucleation Theory assuming only one contact angle. In other words, even when similar ice nucleation properties (one contact angle) are assumed, the slopes become steeper at higher temperature. As a conclusion, the same effect can be explained also by a different hypothesis.*

Good point, the following has been added to the paper.

**'The smaller diversity in the sites active at warmer temperatures may explain the observed steep slopes in $n_s$, however it should be noted that Classical Nucleation Theory also predicts steeper slopes at higher temperatures, assuming a single contact angle.'**

> *p.11 l.19-20 It seems that in Fig.3 the variation might also be explained by the measurement uncertainty? There is no trend in one direction with increasing time, or is the legend incorrectly labelled? This must be clarified.*

The error bars have now been shown more clearly to justify that the overall trend in of both Amelia albite and TUD#3 are not simply an artefact of experimental error. Note that there is some variability for the TUD#3, but the overall trend over 16 months is clear.

> *p.11 l. 25-28 Additionally, the study of Marcolli et al. (2007) and Hartmann et al. (2016) can be mentioned.*

The papers have now been cited.

> *p.13 l.26-32 Larger ns values always imply lower available particle surface area relevant for heterogeneous ice nucleation or lower number of active sites when similar ice nucleating materials are analyzed. The mass concentration and size of a droplet containing particles of different sizes (particle distribution) is not the essential quantity, but the total surface area of the particles. I feel that in both experiments (Atkinson et al., 2013 and Zolles et al., 2015) this quantity is not determined with sufficient reliability. Consequently, this is not a conclusive argument. If ns is carefully derived from the experimental data, this effect should be already considered. Error bars could clarify the uncertainty in ns derived from different experiments.*

This is a good point, we were trying to make clear the differences between the experimental procedures used in different papers but wrote the section poorly. We agree that the total surface area of the particles in a droplet is the important quantity, but wanted to highlight the differences between the experiments. We have improved this discussion with the insertion of:

**"In principle, $n_s$ should be independent of droplet volume and particle concentration, but differences between instruments and methods have been reported (Hiranuma et al., 2015). Additionally, Zolles et al. (2015) estimated the surface area of their feldspar particles using a combination of SEM images and the BET surface area of quartz. This leads to an unspecified uncertainty in their ns values."**

*p.23 Fig.2 For the observed freezing at low temperature, it is necessary to show the limit of the measurement due to pure water freezing.*

We have inserted a new figure showing droplet fractions frozen for all experiments, along with the freezing temperatures for pure water showing that the experiments we have conducted are not interfered with by the background freezing of the instrument, although one of the plagioclase runs gets quite close to this limit.

[Figure]

**Figure 1: Droplet fraction frozen as a function of temperature for 1 wt% suspensions of ground powders of various feldspar samples. The K-feldspars are coloured red, the plagioclase feldspars are coloured blue, the albites are coloured green the feldspar glass is coloured black and the background freezing is coloured cyan. A fit to the**

**background freezing of pure MilliQ water in the µl-NIPI instrument used by Umo et al. (2015) is also included. The shaded area around this fit shows 95% confidence intervals for the fit. It can be seen that all the feldspar samples tested nucleate ice more efficiently than the background freezing of the instrument.**

> *p.23 l.12 I do not agree that ns values can be simply subtracted without introducing further unnecessary uncertainty. An internal mixture of different ice nucleating particles has to be accounted for.*

We do not simply subtract $n_s$ values and have removed the confusing statement. The process is described in Umo et al. (2015). $n_s$ values are not subtracted directly. $K$ values are calculated for the background freezing and subtracted from $K$ values for the experiment. The resulting $K_{het}$ values are then converted into $n_s$ values.

Technical corrections

> *p.3 l.16-18 The citation list in brackets has to be sorted in the order beginning from the oldest to the recent publications. This needs to be changed in the whole manuscript.*

This has been corrected.

> *p.16 l.2 . . . higher temperatures . . .*

A correction has been made.

**References**

[revised manuscript text omitted]

---

## Author Comment (AC2) · 20 Jun 2016

Response to referee 2

We thank the referee for their valuable comments. We have reworked the paper to address the relevant issues where necessary. The reviewer comments are shown in italics, our response in normal type and changes in the manuscript in bold enclosed in inverted commas.

General comments

> *(1) The information provided on the different mineralogical compositions of the feldspar minerals is good, and Figure 1 summarizes this well. I was surprised to not also see more details provided regarding the crystal lattice structure, symmetry, and space group of the different mineral phases. These properties are often referred to in the text to try to understand the observed ice nucleation properties, but without a table or figure summarizing this information it hard to understand this important aspect. Please add as much detail regarding the other known properties of these feldspar minerals in a well organized table or similar.*

We agree that this is valuable information and the crystal lattice structure, point and space groups of the different feldspars has been added to the tables.

> *(2) Introduction: It seems that the recent effort by Perlwitz et al. to incorporate better representations of the variable mineralogical composition of dust into global models should be referred to here. A major challenge regarding understanding and predicting the ice nucleation properties of atmospheric mineral particles is that we do not have a good understanding of the distribution, abundance, and transport of the different mineral types in the atmosphere.*

Perlwitz *et al.* has now been mentioned in the introduction in a new statement:

**'This is an important finding as it has been demonstrated that feldspar is a common component of aerosolised mineral dusts (Glaccum and Prospero, 1980;Kandler et al., 2009;Kandler et al., 2011;Atkinson et al., 2013;Perlwitz et al., 2015)'**

> *(3) On a related note, it is important to also have some discussion of the size of atmospheric feldspar mineral particles. What aerosol size mode are these typically found in? This is crucial to predict their transport, lifetime, and deposition. One of the reasons that the clay minerals have been focused on for so long is that they tend to be present in the smaller atmospheric mineral particle sizes, and thus have longer lifetimes.*

We have added the following text after the comments on the importance of feldspar for ice nucleation:

**'Feldspar particles in the atmosphere tend to be larger than clay particles and so will have shorter lifetimes in the atmosphere, however aerosol modelling work has suggested that feldspar particles can account for many observations of INP concentrations around the world (Atkinson et al., 2013)'**

> *(4) Although you focus on immersion freezing here – without actually stating the heterogeneous ice nucleation mode you measure here (please clarify this so it is clear to the reader), it was odd that this paper on the depositional ice nucleation properties of Feldspar was not cited:*
>
> *Yakobi-Hancock, J. D.; Ladino, L. A.; Abbatt, J. P. D. Feldspar minerals as efficient deposition ice nuclei. Atmos. Chem. Phys. 2013, 13, 11175–11185, doi:10.5194/acp-13-11175-2013.*

We have now made clear in the last paragraph of the introduction and in the experimental section that we make use of heterogeneous immersion freezing:

**'Here this technique is used to make heterogeneous immersion mode nucleation experiments.'**

Also, we have now cited the suggested work in the introduction:

**'Work conducted below water saturation using a continuous flow diffusion chamber has also concluded that feldspars, particularly orthoclase feldspars, nucleate ice at low relative humidity in the deposition mode than other common dust minerals (Yakobi-Hancock et al., 2013).'**

> *(5) Some discussion of the vulnerability of feldspar minerals to chemical attack by e.g. sulfuric acid should be included. This is quite important to understand the actual contribution of feldspar minerals to atmospheric ice nucleation, and also provides some insights into the nature of the ice active sites. Wex et al. (2014), already cited here, discuss some of these aspects. I believe it is well known in the mineralogy community that these feldspar minerals can be readily converted to amorphous clay structure through reaction with sulfuric acid.*

Greater detail into the weathering has been added to both the stability of active sites and the conclusions section to address this issue.

**'This result is in agreement with the fact that albite weathers faster than microcline in soils as $Na^+$ is more readily substituted for hydrogen than $K^+$ (Busenberg and Clemency, 1976; Blum, 1994).'**

In the conclusions:

**'If the high energy defects along exsolution boundaries are responsible for higher ice nucleation activity of K-feldspars then this may offer an insight into acid passivation of ice nucleating ability observed in laboratory studies (Wex et al., 2014). Berner and Holdren (1979) suggest that the acid mediated weathering of feldspar occurs in multiple stages and suggest dissolution of feldspars is concentrated at high surface energy sites such as dislocations and crystal defects, sites which may be related to ice nucleation. More work is needed to explore the significance of exsolution, microtexture and the impact of weathering on feldspars with respect to ice nucleation activity.'**

*(6) Page 6: Why were those 3 mineral samples selected out of the 15 to perform the extended time in water experiments on? It would be valuable to conduct these tests on a larger number of the minerals, since the behavior seems quite variable between minerals. At the least some justification for the 3 sample chosen could be given.*

These experiments were of an opportunistic nature as a rapid decay was noticed in both the Amelia albite and TUD#3 microcline between repeat runs. The justification of these samples being chosen has been added to section 5.2 stability of active sites.

**'TUD #3 microcline and Amelia albite were chosen for this experiment as they contained highly active sites, represented two different types of feldspar and were the only feldspars observed to exhibit this rapid decay in activity. BCS 376 microcline was also included in this activity decay experiment as it had provided consistent data over repeated runs and served as a standard in the Atkinson *et al*. (2013) paper which could therefore be tested.'**

*(7) Page 6, line 31: Citing 8 of the authors own publications that uses the same (and rather simple) experimental method seems like excessive and unnecessary self-citation, especially when Whale et al. (2015a) already provides a detailed discussion of the method. Please restrict these to the most necessary and relevant citations.*

Only the most relevant references have now been cited.

*Page 7, line 19: More of the recently published experimental work that has explored the role of time-dependent freezing should be cited, such as:*

*Wright, T. P.; Petters, M. D. The role of time in heterogeneous freezing nucleation. J.Geophys. Res. Atmos. 2013, 118, 3731–3743, doi:10.1002/jgrd.50365.*

Wright and Petters has now been cited also.

*(8) Some discussion of the similarity of the Monte Carlo approach to estimate the uncertainty of the ns values to other work should be presented. Is this the first time these authors have used this approach, or that anyone has used a similar approach? Wright & Petters (JGR, 2013; cited above) also used a Monte Carlo approach to analyze and interpret their droplet freezing data. Please discuss this. As it is presented it reads as if this is a completely new approach.*

We have inserted the following:

**'Wright and Petters (2013) previously adopted a similar approach to simulate the distribution of active sites in droplet freezing experiments.'**

*(9) Page 8, line 14: "We assume that each droplet contains a representative surface area distribution." Please clarify what "representative" means. Is the goal to account for the non-uniform distribution of particle number and surface area in each droplet?*

We have addressed this issue in response to referee 1. We added:

**'By assuming that the BET surface area of the feldspar powders is made up of monodisperse particles it can be estimated that the droplets will each contain around 106 particles. While there will be a distribution of particle sizes we assume that there are enough particles per droplet that the uncertainty in surface area per droplet due to the distribution of particles through the droplets is negligible. This assumption is supported by our previous work where we show that ns derived from experiments with a range of feldspar concentrations are consistent with one another (Atkinson et al., 2013). If the particles were distributed through the droplets in such a way that some droplet contained a much larger surface area of feldspar than others we would expect the slope of ns with temperature to be artificially shallow. The slope would be artificially shallow because droplets containing more than the average feldspar surface area would tend to freeze at higher temperatures and vice versa. This would mean that ns data derived from experiments with different feldspar concentrations would be inconsistent with one another. However, the fact that ns data for droplets made from suspensions made up with a wide range of different feldspar concentrations all line up shows that the droplet to droplet variability in feldspar surface area is minor (Atkinson et al. 2013). Hence, the droplet to droplet variability in feldspar surface area is neglected and the uncertainty in surface area per droplet in these experiments is estimated from the uncertainties in weighing, pipetting and specific surface area of the feldspars.'**

*(10) Figure 3 is hard to read at the presented size. The symbols are too small and faint.*

We recognise that the figure is hard to read in current form. It was rescaled for the discussion paper. This should hopefully be corrected in the final paper. We have also made an effort to improve the clarity of the error bars.

*(11)    Figure 4 is begging for some error bars or other measurement of the uncertainties, so it can be determined what degree of the observed changes in median freezing temperature are significant and above the experimental uncertainties. It seems that only the Amelia albite sample displayed any significant change.*

We have added temperature error bars to both figures and attempted to improve the clarity of both panels of what is now figure 4.

*(12)    Figure 5: Some annotations/captions added directly to the figure pointing out what data is plotted where would improve the clarity of this paper. It is difficult to have to keep going back to the figure legend to decode what each dataset is from.*

We think that labelling data sets would be quite confusing and add a lot of clutter to the plot. But, we have grouped the like minerals in the key.

Technical corrections

*Page 3, line 19: "experiments"*

This has now been changed.

*Page 7, line 5: Missing a space, should be "C min-1"*

This has been corrected.

*Page 9, line 30: "(nucleation rate) vs." Versus what?*

The typo 'vs' has been removed.

*A space in-between the number and "degreeC" is often missing, such as throughout pages 12 & 13.*

Spaces have been added in the relevant places.

*Page 14, line 18: "regimen this study". Word is missing?*

The text now reads as:

**"Within the microliter regime in this study we have observed some variability amongst the K-feldspars (see Figure 2), but no difference between sanidine and the 4 out of 5 microclines which fall around the line defined by Atkinson et al. (2013)."**

*Page 15, line 13: "sites"*

This has been changed as suggested.

This correction has been made in the text.

**References**

*Atkinson, J. D., Murray, B. J., Woodhouse, M. T., Whale, T. F., Baustian, K. J., Carslaw, K. S., Dobbie, S., O'Sullivan, D., and Malkin, T. L.: The importance of feldspar for ice nucleation by mineral dust in mixed-phase clouds, Nature, 498, 355-358, 10.1038/nature12278, 2013.*

*Berner, R. A., and Holdren, G. R.: Mechanism of feldspar weathering—ii. Observations of feldspars from soils, Geochim. Cosmochim. Acta, 43, 1173-1186, http://dx.doi.org/10.1016/0016-7037(79)90110-8, 1979.*

*Glaccum, R. A., and Prospero, J. M.: Saharan aerosols over the tropical north-atlantic - mineralogy, Mar. Geol., 37, 295-321, 10.1016/0025-3227(80)90107-3, 1980.*

*Kandler, K., Schütz, L., Deutscher, C., Ebert, M., Hofmann, H., Jäckel, S., Jaenicke, R., Knippertz, P., Lieke, K., Massling, A., Petzold, A., Schladitz, A., Weinzierl, B., Wiedensohler, A., Zorn, S., and Weinbruch, S.: Size distribution, mass concentration, chemical and mineralogical composition and derived optical parameters of the boundary layer aerosol at tinfou, morocco, during samum 2006, Tellus, 61B, 32-50, 10.1111/j.1600-0889.2008.00385.x, 2009.*

*Kandler, K., SchÜTz, L., JÄCkel, S., Lieke, K., Emmel, C., MÜLler-Ebert, D., Ebert, M., Scheuvens, D., Schladitz, A., ŠEgviĆ, B., Wiedensohler, A., and Weinbruch, S.: Ground-based off-line aerosol measurements at praia, cape verde, during the saharan mineral dust experiment: Microphysical properties and mineralogy, Tellus, 63B, 459-474, 10.1111/j.1600-0889.2011.00546.x, 2011.*

*Perlwitz, J. P., Pérez García-Pando, C., and Miller, R. L.: Predicting the mineral composition of dust aerosols – part 1: Representing key processes, Atmos. Chem. Phys., 15, 11593-11627, 10.5194/acp-15-11593-2015, 2015.*

*Wex, H., DeMott, P. J., Tobo, Y., Hartmann, S., Rösch, M., Clauss, T., Tomsche, L., Niedermeier, D., and Stratmann, F.: Kaolinite particles as ice nuclei: Learning from*

*the use of different kaolinite samples and different coatings, Atmos. Chem. Phys.,
14, 5529-5546, 10.5194/acp-14-5529-2014, 2014.*

*Yakobi-Hancock, J. D., Ladino, L. A., and Abbatt, J. P. D.: Feldspar minerals as
efficient deposition ice nuclei, Atmos. Chem. Phys., 13, 11175-11185, 10.5194/acp-
13-11175-2013, 2013.*

---

## Author Response (AR2)

Response to referee 1

We would like to thank the reviewer for their helpful comments. We have made some relatively minor
changes to the paper to address the relevant issues. The reviewer comments are written in italics, our
response in normal type and changes to the manuscript in bold.

General comments

*(1) In the new version of the manuscript, the statement that two types of hyperactive feldspar*
*exist which are able to induce freezing at high temperature where usually biological*
*particles, in particular proteins, are presumed to act as ice nucleating particles is weakened*
*(p.11 l.18). But only the influence of biological contaminations is discussed. However, the*
*fact that the sample are only heated to 100°C for 15 minutes do not exclude other heat-*
*resistant species to act as INP at this high temperature. For example in Fukuta 1966 a whole*
*list of organic ice nucleating particles is given showing onset temperatures of -1°C or -1.5°C.*
*I do not doubt that "hyperactive" feldspars exist but I am not convinced that enough test had*
*been made to constrict all possibilities. This still needs to be done and discussed in the*
*manuscript. Further the statements in the abstract (p.1 l27,29) and conclusions (p.17 l.22, 23,*
*24, 30) have to be weakened, too.*

We have changed the relevant sections of the text and cited Fukuta.

In the abstract:

**However, we also show that there is the possibility that some alkali feldspars may have**
**enhanced ice nucleating abilities, which could have implications for prediction of ice nucleating**
**particle concentrations in the atmosphere.**

In the discussion:

**Additionally, it is known that certain organic molecules can nucleate ice efficiently (Fukuta,**
**1966). It is not possible to exclude the possibility of the presence of these or other, unknown heat**
**resistant contaminants that nucleate ice very efficiently.**

In the conclusions:

**We have not excluded the possibility that other entities on the surfaces of the feldspar may be**
**responsible for the ice nucleation observed.**

Specific comments

*p.2 l.4-6:*

*The causal association is not clear. The sentence must be reformulated.*

This now read:

**However, we also show that there is the possibility that some alkali feldspars may have**

**enhanced ice nucleating abilities, which could have implications for prediction of ice nucleating**

**particle concentrations in the atmosphere.**

*p.7 l.21:*

*When the cold stage chamber is flowed with dry nitrogen during the experiment. Do you observe*

*effects from droplet evaporation?*

The droplets evaporate slowly but this has been shown not to impact freezing temperatures. We have added:

**droplets evaporate slowly during experiments however this has been shown to have no**

**detectable effect on freezing temperatures (Whale et al., 2015)**

*p.8 l.7:*

*The meaning of the term "active site" is still not introduced in the paper.*

We disagree with this statement. We have introduced 'active site' on page 8, lines 1 to 13.

*p.10, l21-22:*

*This statement might be true, but applies only for the analysed weight concentration range. Add "for*

*the analysed weight concentration range".*

This has been changed.

*p.12 l.2:*

*I prefer that the authors use always the same word for the same meaning, otherwise the reader thinks about if there is a difference or not. That means the terms "site", "active site", etc. should be unified in the whole manuscript!*

Where appropriate we have added active before site throughout the text.  In some places, the word 'site' is used since it is clear from the context of the sentence that we are referring to.

*p.12 l.4:*

*The term "activation temperatures" is not introduced. What does it mean?*

We have changed this to '**Freezing'** temperatures.

*p.17 l.1:*

*What does "microliter regime" means? This is not physically meaningful. It would be better to give the mass concentration range, number of particles or available surface area, but not the droplet size.*

We have modified this sentence:

**'….surface area regime examined in this study….'**

Technical corrections

*p.7 l.30: Delete "heterogeneous".*

Deleted

*p.7 l.26: "higher" instead of "greater"*

This has been changed

*p.11 l.19; p.17 l.21: "high" instead of "warm"*

This has been changed

*p.11 l.28: "lower" instead of "colder"*

This has been changed

*p.12 l.18:*

*Delete unnecessary information: "the time between successive runs".*

This has been changed

*p.17 l.20 and l.25:*

*Delete "generic".*

This has been changed

*p.17 l.23 and l.24:*

*Use consistent notation for "hyperactive".*

This has been changed

*Fig. 4:*

*The Figure has a bad resolution.*

This will be resolved in the published version.

**References**

[revised manuscript text omitted]